# How Severe Anaemia Might Influence the Risk of Invasive Bacterial Infections in African Children

**DOI:** 10.3390/ijms21186976

**Published:** 2020-09-22

**Authors:** Kelvin M. Abuga, John Muthii Muriuki, Thomas N. Williams, Sarah H. Atkinson

**Affiliations:** 1Kenya Medical Research Institute (KEMRI) Center for Geographical Medicine Research-Coast, KEMRI-Wellcome Trust Research Programme, Kilifi P.O. Box 230-80108, Kenya; jmuriuki@kemri-wellcome.org (J.M.M.); twilliams@kemri-wellcome.org (T.N.W.); 2Centre for Tropical Medicine and Global Health, Nuffield Department of Medicine, University of Oxford, Oxford OX3 7FZ, UK; 3Department of Infectious Diseases and Institute of Global Health Innovation, Imperial College, London W2 1NY, UK; 4Department of Paediatrics, University of Oxford, Oxford OX3 9DU, UK

**Keywords:** severe anaemia, bacteraemia, iron, hepcidin, *Salmonella*, *E. coli*, *Staphylococcus*, *Haemophilus*, *Streptococcus*, Mendelian randomization

## Abstract

Severe anaemia and invasive bacterial infections are common causes of childhood sickness and death in sub-Saharan Africa. Accumulating evidence suggests that severely anaemic African children may have a higher risk of invasive bacterial infections. However, the mechanisms underlying this association remain poorly described. Severe anaemia is characterized by increased haemolysis, erythropoietic drive, gut permeability, and disruption of immune regulatory systems. These pathways are associated with dysregulation of iron homeostasis, including the downregulation of the hepatic hormone hepcidin. Increased haemolysis and low hepcidin levels potentially increase plasma, tissue and intracellular iron levels. Pathogenic bacteria require iron and/or haem to proliferate and have evolved numerous strategies to acquire labile and protein-bound iron/haem. In this review, we discuss how severe anaemia may mediate the risk of invasive bacterial infections through dysregulation of hepcidin and/or iron homeostasis, and potential studies that could be conducted to test this hypothesis.

## 1. Introduction

Child mortality due to preventable causes remains disproportionately high in sub-Saharan Africa [1]. Severe anaemia is prevalent in African children [2,3,4], and accounts for 6–28% of all hospital admissions with case fatality rates of 4–10% [5,6,7]. The causes of severe anaemia are multiple, and often coexist, including infections, haemoglobinopathies, and nutritional deficiencies [8,9]. At the same time, invasive bacterial infections account for 6–15% of febrile hospital admissions with case fatality rates of 5–28% [10,11,12,13]. The commonest bacterial isolates observed in African children are *Streptococcus pneumoniae*, *Staphylococcus aureus*, non-typhoidal *Salmonellae* (NTS), *Haemophilus influenzae,* and *Escherichia coli* [10,11,12].

Severe anaemia aetiologies can be grouped into hyporegenerative (anaemias due to iron and other nutritional deficiencies, pure red cell aplasia, anaemia of inflammation, aplastic anaemia, erythropoietin underproduction, and marrow infiltration), and regenerative (anaemias due to haemolysis, immune dysregulation, haemorrhage, and non-immune factors (haemoglobinopathies, drugs, microangiopathy, and hypersplenism)) [14]. Absolute iron deficiency (defined as low serum iron, low ferritin, and elevated transferrin iron binding capacity) results from blood loss, increased physiological demands of iron, intake of staple foods with low iron bioavailability, and malabsorption. The diagnosis of iron deficiency anaemia is usually based on observation of microcytic red cell features, although haemoglobin E/C and alpha/beta thalassaemia need to be ruled out. Iron deficiency is a common cause of anaemia in sub-Saharan Africa [4], but is less frequently observed in children with severe anaemia [8,9,15]. Deficiencies of vitamin A and vitamin B12 (cobalamin) are also common in African children [9], and are associated with severe anaemia [9,15]. In sickle cell disease, anaemia is secondary to haemoglobin polymerization leading to red blood cell deformation and lysis. Sickle cell disease may also induce iron deficiency anaemia through increased iron utilization to replace damaged red blood cells or urinary iron loss [16]. Anaemia of inflammation (low serum iron and normal/high ferritin levels) is found in patients with infections (parasitic, bacterial, viral, and fungal), cancer, or autoimmune disorders [14,17], and is thought to be induced by the hepatic iron regulatory hormone hepcidin [18]. While persistently raised hepcidin levels may protect against invading pathogens [19,20], enhanced iron sequestration may increase susceptibility to intracellular infections [21]. Regenerative anaemias are characterized by high reticulocyte counts due to increased haemolysis or haemorrhage [14]. In sub-Saharan Africa, little is known about the putative aetiologies of severe anaemia from a public health perspective. Nonetheless, most of these aetiologies are important, often coexist in a single patient [9,15], and may contribute to a risk of infection either individually or synergistically.

Epidemiological studies have found strong associations between severe anaemia and invasive bacterial infections [9,22,23], in particular with NTS bacteraemia [24,25,26]. Individually, some of the common causes of severe anaemia have been associated with an increased risk of invasive bacterial infections in African children including severe malaria [27,28,29], malnutrition [11], human immunodeficiency virus (HIV) [11,27], and sickle cell disease [30,31]. This may be due to several interlinking pathways including increased haemolysis and erythropoietic drive, immune dysfunction, and gut permeability (Figure 1). Chronic haemolysis, haemolytic crisis, and increased erythrophagocytosis are common features of haemoglobinopathies and infections such as malaria [16,32]. In murine models, severe anaemia caused by haemolytic mechanisms (chemical, parasitic or antibody-mediated), but not by phlebotomy, increases susceptibility to invasive bacterial infections [33,34,35,36]. The pathophysiology of haemolytic anaemia is complex. One critical feature is tissue iron overload and increased serum iron that exceeds the transferrin binding capacity. Additionally, the production of the hepatic iron-regulatory and antimicrobial hormone hepcidin is downregulated during severe anaemia due to the action of erythroferrone (ERFE) [37,38,39,40].

In this review, we outline the hypothesis that severe anaemia contributes to the burden of invasive bacterial infections in African children through the disruption of iron homeostasis and/or iron-regulatory proteins.

## 2. The Severe Anaemia—Bacteraemia Hypothesis

We hypothesize that iron dysregulation is at the centre of the association between severe anaemia and invasive bacterial infections. Iron is an essential micronutrient for all living organisms. The unique ability of iron to serve as an electron donor and acceptor renders it critical for various cellular and immune processes including nucleic acid synthesis, cellular proliferation, mitochondrial respiration, and generation of microbicidal reactive oxygen species (ROS) [41]. Nonetheless, excessive iron is toxic because harmful hydroxyl radicals are generated through the Fenton reaction that can dampen the effector functions of mononuclear cells [41]. Due to its toxicity and biological significance, iron homeostasis is tightly regulated. In circulation, the net iron concentration is maintained through efficient macrophage-recycling of iron from senescent or damaged red blood cells, and the effective use of iron in the bone marrow for erythropoiesis. Most intracellular iron is complexed to haem or the iron-storage protein ferritin, while extracellular iron is bound to high-affinity chaperone proteins including transferrin, haptoglobin, hemopexin, lipocalin-2, and lactoferrin. The hepatic hormone hepcidin is the master iron regulator, and maintains iron homeostasis by controlling the absorption of dietary iron, release of iron from storage cells, and sequestration of recycled iron in macrophages [42]. Infections by extracellular pathogens result in cellular iron import via various receptors including those of transferrin, lipocalin-2, haem-haemopexin (CD91), and haemoglobin-haptoglobin (CD163) complexes. Elevated hepcidin levels further ensure that iron is maintained intracellularly by degrading the sole iron exporter, ferroportin [43]. The reduced availability of iron in plasma “starves” invading pathogens and protects against extracellular infection. Low hepcidin levels, observed during severe anaemia [37,38,39,40], may undermine this nutritional immunity. Intracellular infections, on the other hand, are associated with reduction in cellular haem-iron content and rely on suppression of iron import into macrophages and/or increased iron export out of cells. An additional defence strategy involves iron export out of phagolysosomes, such as the *Salmonella*-containing vacuole (SCV), using the natural resistance-associated macrophage protein 1 (Nramp1). Concomitant infections that promote iron sequestration into macrophages may disrupt these iron regulation strategies [21,35], and predispose African children with severe anaemia to an increased risk of intracellular bacterial infections.

Although iron deficiency is common in sub-Saharan Africa [4], epidemiological studies have found either no association between iron deficiency and severe anaemia [8], or that iron deficient children living in areas of very high infectious burden are less likely to be severely anaemic [9,44]. Infections and haemoglobinopathies are strongly associated with severe anaemia [7,9], and malaria may be the commonest cause of severe anaemia among children in sub-Saharan Africa [7,45,46,47]. The link between infections, haemoglobinopathies, and iron deficiency is complex [30,32]. In African children, iron deficiency has been reported to protect against infections such as malaria [48,49]. Unlike other aetiologies of severe anaemia [27,28,29,30,31], there is limited data on the risk of invasive bacterial infections in children with absolute iron deficiency. While iron deficiency may restrict bacterial growth due to limited iron availability [50], very low iron levels may also negatively impact immune responses to invasive bacterial infections [51,52]. Iron is required for the development and effector functions of immune cells, including proliferation of T cells and formation of ROS through the Fenton reaction [51,53,54]. Severe anaemia and iron deficiency also strongly downregulate the antimicrobial hormone hepcidin even during infection thus facilitating the release of iron from storage cells and subsequent loss of hepcidin-induced “hypoferraemia of infection” and “nutritional immunity” [37,55]. In a population-based survey of Norwegian adults, iron deficiency was associated with increased risk of bloodstream infections [56], possibly through immune dysregulation [57]. The study did not report some measurements including haemoglobin, hepcidin, and ferritin levels, which limits how the data may be interpreted in the context of severe anaemia. On the other hand, chronic infection may induce functional iron deficiency. Such “hypoferraemia of infection” is a host defence strategy to limit iron availability from invading pathogens [58], and involves sustained production of proinflammatory cytokines such as interferon-gamma (IFN-γ), tumour necrosis factor-alpha (TNF-α), interleukin (IL)-1 and IL-6 [17]. These proinflammatory cytokines can exacerbate existing severe anaemia by promoting dyserythropoiesis, iron sequestration, and erythrophagocytosis [17,59]. IL-6 promotes the production of hepcidin [18], and has been reported to be upregulated in children with severe malarial anaemia [60,61]. However, ERFE may have a stronger negative effect on hepcidin production during severe anaemia [39], and low hepcidin levels have been observed in anaemic children and young women with concomitant inflammation [37,62]. Whether and how IL-6 or other proinflammatory cytokines may increase iron sequestration independently of hepcidin in severely anaemic children remain unknown.

The tight junctions between gut epithelial cells form a barrier that is normally impermeable to enteric pathogens. However, African children with severe anaemia have a high risk of bacteraemia due to enteric organisms, particularly NTS and *E. coli* [23,24,25,26], and this has been observed in severely anaemic children generally [9] and in those with underlying sickle cell disease [30], malaria [27,63], HIV [27,64], and malnutrition [65,66]. Severe anaemia may promote gut permeability through persistent intestinal inflammation, immune dysfunction, and gut dysbiosis (Figure 1). Children with low haemoglobin levels and those receiving iron supplementation have been reported to have dysbiosis of gut microbiota and increased presence of pathogenic bacteria in the gut [67,68,69]. An in vitro study using enterocyte-like Caco-2 cells found increased invasiveness and survival of *Salmonella enteritidis* when iron concentrations were increased as might be expected in acute haemolysis [70]. In model studies, neonatal mice with severe anaemia had a persistent increase in intestinal permeability and electron micrographs showed abnormalities of epithelial adherens junctions probably due to destabilization of the E-cadherin mRNA [71] or decreased expression of the tight junction protein zonula occludens-1 (ZO-1) [72]. Phlebotomy-induced severe anaemia was associated with increased intestinal mucosal hypoxia and production of IFN-γ by intestinal macrophages, which may contribute to increased gut permeability and development of necrotizing enterocolitis [72]. Depletion of intestinal macrophages ablated the effects of severe anaemia on the intestinal barrier activity. Whether other immune cells, including neutrophils [73], also contribute to necrotizing enterocolitis remains unknown. Gut inflammation may also directly induce dysbiosis, through the actions of ROS, calprotectin, and other inflammatory mediators [74,75,76]. Further studies are required to elucidate the precise mechanisms of gut dysbiosis during severe anaemia, inflammation-mediated gut permeability, and how this influences the development of systemic infections.

Severe anaemia may also promote invasive bacterial infections through modulation of immune responses. Protection against invasive bacterial infection relies on a coordinated and regulated innate and adaptive immune response. In the initial stages of infection, local macrophages engulf and destroy the invading pathogens, and produce cytokines and chemokines to induce an inflammatory response. Monocytes and neutrophils are rapidly recruited to the site of infection. These cells destroy phagocytosed organisms effectively using the NADPH oxidase-dependent ROS production [77,78]. Neutrophils can also destroy extracellular organisms through degranulation, releasing of neutrophil-extracellular traps (NETs), and/or production of ROS [77,79]. Severe anaemia due to iron deficiency may promote impaired development and apoptosis of immune cells [51,53,54], including neutrophil hypersegmentation [80] and impaired neutrophil/monocyte oxidative burst [81,82]. There is limited clinical research regarding the effect of severe anaemia on white blood cell differential count. In some case reports, iron deficiency anaemia has been associated with neutropenia [83,84]. The effects on neutrophils are supported by murine models of severe malarial anaemia, which reported reduced neutrophil influx and lower proinflammatory cytokine (IL-17 and IFN-γ) levels [85,86], and higher levels of the anti-inflammatory cytokine IL-10 (IL-10) [34,87]. Elevated IL-6 levels have been observed in field studies of children with severe malaria anaemia [60,61] and IL-6 may be upregulated when hepcidin levels are low [88] further inhibiting neutrophil influx [89]. IL-6 is an important checkpoint regulator of neutrophil trafficking and promotes clearance of neutrophils and recruitment of monocytes [89,90]. Downregulation of neutrophil responses may contribute to poor clearance of invasive bacterial pathogens in children with severe anaemia.

Haemolytic anaemias induce sustained release of free haem and non-transferrin-bound iron (NTBI), which may impair the recruitment and function of myeloid cells [33,91]. Free haem dampens the ability of phagocytes to kill ingested bacteria, including through reduced neutrophil mobilisation and oxygen burst activity [33,92]. Haem was also found to disrupt the actin cytoskeleton rearrangement, which is crucial for recruitment and migration of phagocytic cells [91]. This may ultimately cause immune paralysis and impede resistance to bacterial infections. The haem-catabolizing enzyme, haem oxygenase-1 (HO-1), may also downregulate immune responses to bacterial infections. HO-1 is normally expressed at low levels in most tissues but is highly induced by inflammation, hypoxia, and other stimuli. In conditions with increased haemolysis, HO-1 is induced to break down elevated free haem into equimolar amounts of carbon monoxide, biliverdin, and ferrous iron. Whilst its induction reduces oxidative damage by free haem, HO-1 is associated with reduced elimination of pathogens including systemic NTS, malaria, and leishmaniasis [33,92,93]. This may be a result of the direct tolerogenic effects of HO-1 on the immune system [94], or due to the actions of its products. Biliverdin and carbon monoxide are anti-inflammatory and scavenge radical molecules that kill intracellular bacteria [92,95]. Moreover, intracellular iron inhibits the activity of IFN-γ in a dose-dependent manner [96,97]. IFN-γ is central to the control of intracellular pathogens by inducing ROS generation through the nitric oxide synthase pathway [98]. Inhibiting the expression of IFN-γ increases the availability of iron for intracellular pathogens by increasing the uptake of transferrin-bound iron into macrophages [99] and storage of iron in ferritin [100]. Iron may also inhibit the expression of other inflammatory mediators including tumour-necrosis factor and nitric oxide synthase [101,102].

Invasive bacteria similarly require iron for their metabolic and pathogenic processes. As such, bacteria have evolved multiple, and often redundant, strategies to acquire ferrous (Fe^2+^), ferric (Fe^3^), and/or haem-containing proteins from their host. These strategies are varied depending on the lifestyle (intracellular or extracellular) and preferred iron source (intracellular labile iron, protein-bound iron/haem, or NTBI) of the organism and include: (1) siderophore or haemophore production, (2) breakdown of ferroproteins including haem and haemoglobin, (3) ferric and ferrous iron uptake systems, and (4) transferrin, lactoferrin, and ferritin receptors. Siderophores are high-affinity low-molecular-weight iron chelators that can extract ferric iron from human chaperone proteins including transferrin, and which are actively transported across bacterial membranes [103]. Severe anaemia and haemolysis increase circulatory and tissue iron concentrations, making direct and siderophore-mediated iron acquisition easier for invading pathogens (Figure 2). Additionally, low hepcidin levels, observed during severe anaemia [37,39], promote increased dietary iron absorption, and release of iron stored in macrophages [42].

Below we discuss iron acquisition strategies of the pathogenic bacteria that are commonly isolated in blood cultures of African children [10,11,12], and how severe anaemia might influence iron availability for these organisms.

### 2.1. Gram-Negative Organisms

Gram-negative bacteria have developed complex and redundant systems to acquire iron and/or haem from a wide range of molecules. These include secretion of high-affinity siderophores or direct binding of haem/haemoproteins, lactoferrin, or transferrin by substrate-specific receptors in the outer membrane. The siderophore or iron-containing proteins are actively transported into the periplasmic space, a process that is TonB dependent [104]. Subsequently, iron or haem is transported across the inner membrane using binding-protein-dependent ATP-binding cassette (ABC) systems or the universal Feo system. Gram-negative bacteria often have multiple iron transporters that utilize different substrates. However, the preferred iron sources and transporter systems are highly bacteria specific.

#### 2.1.1. Non-Typhoidal *Salmonellae*

NTS are a group of Gram-negative macrophage-tropic intracellular bacteria. NTS bacteraemia is strongly associated with severe anaemia in African children [24,25,26]. NTS acquire iron through high-affinity siderophores (salmochelin and enterobactin), capture of transferrin iron, and uptake of Fe^2+^ [105,106,107]. In model studies, *Salmonella* was reported to preferentially infect haemophagocytic macrophages [108] or impaired granulocytes [33], where it acquires iron for its growth and proliferation [109]. The regulation of intracellular iron by hepcidin and its ligand ferroportin is an important determinant of NTS survival [110,111]. Ferroportin is present on the phagolysosomal membrane forming the SCV [112,113], and may transport iron into the SCV from the cytoplasm [113]. Model studies using exogenous hepcidin have reported an increase in *Salmonella* proliferation, [100,114] probably due to increased iron sequestration [21]. On the other hand, increased risk of invasive *Salmonella* infection has been observed in *HAMP* (hepcidin gene) knockout studies [115], suggesting that the low hepcidin levels reported in children with severe anaemia [37,116] may contribute to NTS susceptibility. Nonetheless, significant gaps remain in our understanding of the implications of iron uptake through ferroportin on the SCV including: (1) Whether it promotes bacterial growth by providing an essential nutrient or bacterial clearance by generating ROS through the Fenton reaction; and (2) whether it works synergistically or antagonistically with Nramp1, which transports divalent metals out of phagolysosomes.

#### 2.1.2. *Escherichia coli*

*E. coli* are Gram-negative extracellular bacteria that inhabit the gastrointestinal tract as commensals but can also cause invasive disease and sepsis. *E. coli* bacteria acquire environmental iron using high affinity siderophores (enterobactin, salmochelin, aerobactin, and yersiniabactin), the ferrous iron (Feo) uptake system, and/or haem receptors [117,118,119]. The uptake of ferrous iron by *E. coli* requires the reduction of extracellular ferric iron, and this is mediated by ferric reductases [120]. Studies using serum from iron-supplemented individuals [121] or hypoferraemic cord blood [122] reported that the growth rates of *E. coli* were substantially increased in the presence of iron. In murine models, *E. coli* was reported to utilize NTBI [119], and its growth is repressed by exogenous hepcidin [114,119,123]. European children given intramuscular iron dextran [124], patients with aplastic anaemia [125], and an adult with iron overload [126] had increased susceptibility to *E. coli* sepsis. Consequently, it is plausible that increased serum and tissue iron levels during haemolysis and severe anaemia may similarly increase the risk of *E. coli* infections. In model studies, iron deficiency is associated with protection against *E. coli* sepsis [50,119], however severe iron deficiency anaemia reduces hepcidin production even during inflammation [37,127], and very low hepcidin levels have been associated with increased susceptibility to *E. coli* infections [119,128]. Iron deficiency may also be associated with poor immune development and responses to infections [51,53,54]. Additionally, severe anaemia is associated with increased gut permeability [71], which increases the risk of sepsis by gut pathogens such as *E. coli*. Further studies are required to decipher precise iron acquisition strategies employed by *E. coli* during systemic infections and the role of severe anaemia and/or haemolysis in the pathogenicity of *E. coli* in African children.

#### 2.1.3. *Haemophilus influenzae*

*H. influenzae* are fastidious Gram-negative bacteria that inhabit the human nasopharynx as commensals. *H. influenzae* cannot synthesize protoporphyrin IX, and rely on exogenous haem or iron in the presence of protopophyrin IX to grow [129]. As a result, *H. influenzae* encode several receptors that bind host haemoproteins, including the haem- (HbpA), haemoglobin- and haemoglobin-haptoglobin- (HgpA, HgpB and HgpC) binding proteins [130,131,132]. *H. influenzae* also secrete haemophores (HxuA) that capture free haem or haem complexed to haemopexin [133]. Additionally, *H. influenzae* can acquire transferrin-bound iron through the periplasmic ferric-ion binding protein, FbpA [134]. In observational studies, *H. influenzae* has been reported to be associated with haemolytic anaemia [135] and severe anaemia [136] in hospitalized children. In these studies, the development of anaemia was observed over the course of the infection. It is plausible that haemolytic factors secreted by the bacteria to aid acquisition of haem might contribute to the development of anaemia. The reverse could also be true, where underlying haemolytic conditions promote growth and proliferation of *H. influenzae*. Indeed, African children with sickle cell disease and HIV, which induce chronic haemolytic anaemias [137,138], have an increased risk of *H. influenzae* infections [11,30].

#### 2.1.4. Other Gram-Negative Organisms

Siderophilic bacteria, including *Yersinia enterocolitica* 09, *Vibrio vulnificus,* and a pneumonia model of *Klebsiella pneumoniae,* utilize NTBI in iron overload conditions, and hepcidin-mediated hypoferraemia protects against these pathogens [19,139,140]. This NTBI uptake may be facilitated by the universal Feo system, which promotes the uptake of environmental iron in Gram-negative bacteria [104,141]. *Yersinia enterocolitica* also utilizes the hemin receptor (HemR) to obtain haem-bound iron [142]. *Neisseria* spp. encode a set of transferrin-binding proteins (TbpA and TbpB), lactoferrin-binding proteins (LbpA and LbpB), and haemoglobin receptors (HmbR) [143,144]. Intracellular *Francisella tularensis* can acquire cytoplasmic transferrin-bound iron using transferrin receptors [106]. Severe anaemia results in increased circulating and intracellular iron and haem levels needed by these bacteria. Further work is needed to determine how haem, iron, and hepcidin dysregulation in severe anaemia influence the proliferation and bactericidal activity of these organisms.

### 2.2. Gram-Positive Organisms

The iron acquisition strategies of Gram-positive bacteria are generally based on binding-protein-dependent ABC transporter systems. They include cell surface proteins anchored to the membrane, which bind Fe^2+^, Fe^3+^, and haem-containing compounds and transport them using ABC permeases [104]. Various Gram-positive bacteria employ additional iron acquisition strategies. *Bacillus anthracis* utilizes two secreted haemophores, IsdX1 and IsdX2, which bind free haem or extract haem from haemoproteins [145]. On the other hand, *Bacillus cereus* can use the near-iron transporter (NEAT) domain protein to bind plasma ferritin and haemoglobin [146]. Nevertheless, little is known about the implications of severe anaemia on the iron and haem acquisition systems of Gram-positive bacteria.

#### 2.2.1. *Staphylococcus aureus*

*S. aureus* are Gram-positive extracellular organisms that normally colonize the anterior nares and skin. Invasive *S. aureus* infections can cause numerous local, bloodstream, and tissue infections. The preferred iron source for *S. aureus* is haem, whose uptake is mediated by the iron-regulated surface determinant (Isd) system [147,148]. Interestingly, high exogenous haem concentrations were reported to be toxic to *S. aureus* in in vitro studies [149]. However, the authors found that *S. aureus* can sense and adapt to haem toxicity when the bacteria are first cultured in a medium containing sub-toxic haem concentrations. *S. aureus* can also acquire iron from transferrin either directly [150], or using two siderophores (staphyloferrin A, encoded by the *sfaABCD* gene cluster, and staphyloferrin B, encoded by the *sbnABCDEFGHI* operon) [151,152]. However, there is conflicting information on the effect of hepcidin and extracellular iron concentrations on the growth and proliferation of *S. aureus*. In vitro growth rates of *S. aureus* were substantially reduced in media supplemented with hepcidin [123] or in hypoferraemic cord blood [122]. Nonetheless, the impact of hepcidin or hypoferraemic cord blood on the growth rates of *S. aureus* was substantially lower than for *E. coli*. Conversely, studies using murine models [139] and serum from iron-supplemented individuals [121] found no effect of hepcidin or NTBI on the growth rates of *S. aureus*. Further studies are required to ascertain how *S. aureus* acquires iron in severely anaemic children, and the effects of very low haemoglobin levels and/or increased haemolysis on the growth rates of *S. aureus*.

#### 2.2.2. *Streptococcus pneumoniae*

*S. pneumoniae* are Gram-positive extracellular bacteria normally colonizing the respiratory tract, but which can cause invasive and local infections. In epidemiological studies, pneumococci are strongly associated with sickle cell disease [30,31] and HIV [11]. Relatively little is known about pneumococci’s iron acquisition strategies. Growth of pneumococci in iron-restricted in vitro media is supported by Fe^2+^, Fe^3+^, and haem-containing compounds, but not by human ferritin, holo-transferrin, or holo-lactoferrin [153,154,155]. There are no known pneumococcal siderophores or haemophores. Iron uptake by pneumococci is facilitated by three ABC transporter operons—*piu*, *pia,* and *pit* [156]. Pneumococcal haem uptake is mediated by the haemoglobin-binding membrane proteins Spbhp-22 and Spbhp-37 [154]. It is conceivable that severe anaemia increases the risk of pneumococci through increased NTBI and free haem. In vitro studies report an upregulation in hepcidin expression upon infection with *S. pneumoniae*, which is mediated by inflammatory stimuli [157]. However, there is limited knowledge on how the hepcidin-mediated activity influences the growth and virulence of pneumococci.

## 3. Testing the Hypothesis

### 3.1. In Vitro and Animal Model Studies

In vitro studies have contributed greatly to the understanding of bacterial iron acquisition mechanisms under artificially controlled conditions. Ex vivo studies, which use human blood or tissues for investigation, offer minimal alteration of natural conditions. In vitro and ex vivo models of monitoring bacterial growth in different concentrations of iron or human ferroproteins have been established for some bacteria [122,152,158]. These studies offer an advantage, as they are easy to perform and allow quick evaluation of bacterial responses to different physiological conditions created by severe anaemia. However, severe anaemia involves a complex web of iron and immune regulatory mechanisms that may not be captured in in vitro or ex vivo studies. It may also be difficult to establish the consequences of such long-term exposures to anaemia aetiologies on bacterial growth.

Laboratory animals can be manipulated to mimic severe anaemia phenotypes, including haemolysis, genetic predisposition, haemoglobin-depletion, and iron deficiency anaemia, and susceptibility to different bacterial organisms can be measured. Murine studies also allow the evaluation of inflammatory and iron biomarker responses to bacteraemia and severe anaemia. The use of murine genetic knockout models enables estimation of the significance of particular proteins or sets of genes in susceptibility to invasive bacterial infections during severe anaemia, and have been widely used in hepcidin and ferritin studies [100,115]. Extrapolating findings from such studies can enable basic understanding of processes that promote susceptibility to bacterial infections in children with severe anaemia and can inform preliminary formulation of therapeutic approaches. Nonetheless, laboratory mice do not have the same genetic diversity, which influences immune responses to bacterial infection, as the human population. Additionally, mice and human iron- and haem-containing molecules may not have similar properties, and this may influence the inferences made from such studies. For example, bacterial iron acquisition proteins, such as IsdB for *S. aureus*, have been reported to bind more readily to human than to mice haemoglobin [159].

### 3.2. Observational Studies

Cross-sectional and case-control studies have been used to study associations between severe anaemia and invasive bacterial infections in African children. Such studies have provided information on the strong association between severe anaemia and invasive bacterial infections, and particularly with NTS in African children. However, such observational studies are marked with inherent challenges including reverse causality, sampling biases, and unmeasured confounding. Invasive bacterial infections may be predisposing factors for severe anaemia. Underlying conditions such as malaria, malnutrition, and HIV infections, which are associated with both severe anaemia and invasive bacterial infections [11,27,160] are common in African children and are important confounders. The poor sensitivity of blood cultures may also contribute to sampling errors in such studies.

### 3.3. Mendelian Randomization

Randomized control trials (RCTs) are the “gold standard” for studying causal associations between an exposure and an outcome. However, the severity and complexity of severe anaemia makes it challenging to randomize children for treatment while investigating bacterial susceptibility. Mendelian randomization (MR) studies utilize genetic variants, which act as instrumental proxies for modifiable exposures (e.g., severe anaemia), to infer their causal relationship with health outcomes (invasive bacterial infections) [161]. Similar to RCTs, MR studies are based on a random allocation of groups based on Mendel’s laws of random segregation and independent assortment. This reduces biases of environmental confounding and reverse causation, which regularly influence observational studies. The validity of MR studies is based on three principle assumptions. The instrumental variable (1) must be reliably associated with the health outcome, i.e., invasive bacterial infections; (2) should not be associated with potential confounders; and (3) should influence the outcome only through the exposure of interest, that is, the genetic variant should influence invasive bacterial infections only through severe anaemia causal pathways [161,162]. As discussed earlier, severe anaemia has multiple causes with different clinical profiles. Consequently, bacterial susceptibility pathways may be different with each underlying cause of severe anaemia. Additional polymorphisms associated with risk of or protection from severe anaemia should be studied in relation to invasive bacterial infections. However, this approach may also have limitations, including the identification of plausible genetic variants and the very large sample sizes required.

## 4. Perspectives and Implications of the Hypothesis

Childhood iron supplementation is recommended in regions in which the prevalence of anaemia is high [163]. In such regions, the incidence of both severe anaemia [3,8], and invasive bacterial infections [10,11,12,13] is also high. If severe anaemia increases the risk of invasive bacterial infections through haemolysis and iron dysregulation, then iron supplementation may exacerbate adverse outcomes. The identification of organisms whose susceptibility is mediated by iron dysregulation, and the precise mechanisms of such susceptibility, should be a research priority. Mechanisms to minimize the risk of invasive bacterial infections by targeting iron regulatory pathways represent an exciting novel therapeutic target, and would further boost efforts to reduce preventable deaths in African children.

## Figures and Tables

**Figure 1 ijms-21-06976-f001:**
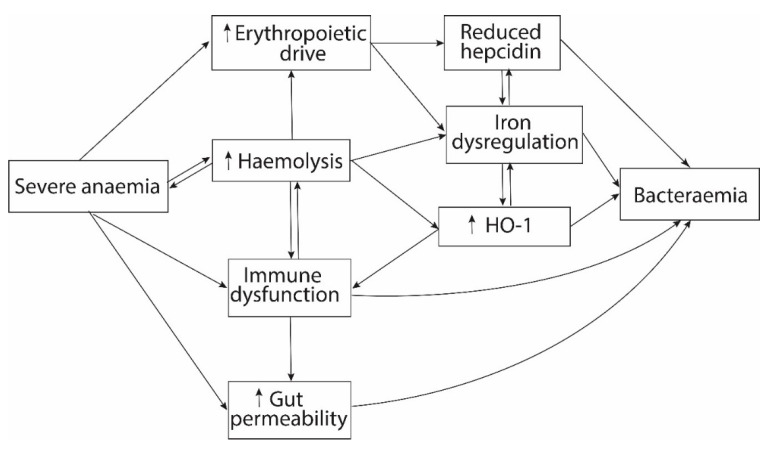
The link between severe anaemia and bacteraemia. Severe anaemia may increase the risk of invasive bacterial infections through several interlinking pathways including increased erythropoietic drive, haemolysis, immune dysfunction, and gut permeability. In both haemolytic and non-haemolytic severe anaemia, elevated erythropoietic drive increases erythroferrone levels, reducing hepcidin, and altering macrophage iron sequestration. This increases iron availability for invading bacterial pathogens. Haemolysis increases the levels of non-transferrin-bound iron, free haem, and haem oxygenase-1 (HO-1), which are associated with immune dysfunction and dysregulation of iron homeostasis.

**Figure 2 ijms-21-06976-f002:**
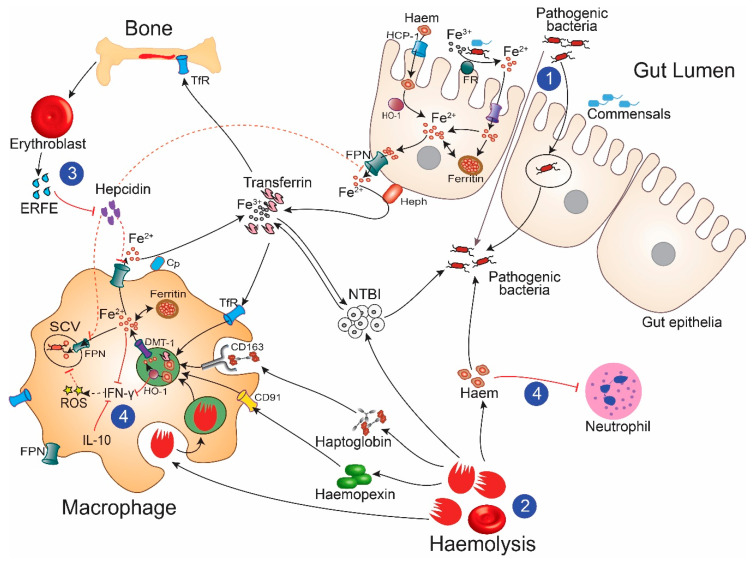
The severe anaemia-bacteraemia hypothesis. Children with severe anaemia may have increased gut permeability promoting the invasion of pathogenic bacteria from the gut lumen (1); haemolysis, which increases the availability of haem and non-transferrin bound iron (NTBI) for extracellular and intracellular organisms (2); increased erythropoietic drive, which inhibits the antimicrobial hepcidin allowing increased availability of iron for extracellular bacteria and movement of iron into the *Salmonella* containing vacuole (SCV) (3); and immune dysregulation including the inhibition of recruitment and effector function of immune cells such as neutrophils or production of pro-inflammatory cytokines such as interferon-gamma (IFN-γ) (4). TfR denotes transferrin receptor; ERFE: Erythroferrone; HO-1: Haem oxygenase-1; Cp: Ceruloplasmin; ROS: Reactive oxygen species; FPN: Ferroportin; IL-10: Interleukin-10; FR: Ferric reductase; Heph: Hephaestin; HCP-1: Haem carrier protein-1; and DMT-1: Divalent metal transporter-1. Black arrows indicate increased activity; red arrows, inhibitory pathways; and dotted lines, suppressed activity.

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
