# Peer review of "How Severe Anaemia Might Influence the Risk of Invasive Bacterial Infections in African Children"

_ijms, 2020, doi:10.3390/ijms21186976_

Round 1

Reviewer 1 Report

The manuscript intitled "How severe anemia might influence the risk of invasive bacterial infections in African children" submitted as an "Hypothesis" paper by Abuga K.M. et al is clear and well written. However, the main mecanism explaining the probable link between infection and anemia in African children is haemolysis in the manuscript.

Major Comment:

In the paper of kassebaum et al published in Blood (2014 123:615-624), "a systematic analysis of global anemia burden from 1990 to 2010", the main cause of anemia in Africa is iron deficiency (prevalence around 20 000/100 000, figure 3) were no mecanism of haemolysis is observed.The second is malaria and the third is shistosomiasis. Anemia linked to Sickle cell disease or Thalassemia represent less than 5000/100 000  in 2010. Regarding this figure, more than 1/3 of the anemia is linked to iron deficiency where no Haemolysis is observed.

The author should add a specific paragraph in the introduction to disccuss the mecanism of infection when the iron and ferritine levels are low and not available for the pathogenic bacteria.

Moreover, in the inflamatory process, Haemolysis is not the main mecanism leading to anemia.

The authors should focus on the hemolytic anemia only, like in sickle cell, thallassemia or malaria for example.

Minor comments:

-In the third paragraph (ligne 269), the subpart should be 3.1 and not "a)" etc..

- the author should precised  that rs334 is a mutation in the beta globin gene.

- Another hypothesis should be take into account. Indeed, the  leucocyte count, and more specifically the neutrophils count should be of interest. For example, not only hemoglobin level but a CBC + DIFF  overview is mandatory to explain infection. The neutrophils are key cells to struggle with pathogenic bacteria.

Author Response

Reviewer #1

The manuscript intitled "How severe anemia might influence the risk of invasive bacterial infections in African children" submitted as an "Hypothesis" paper by Abuga K.M. et al is clear and well written. However, the main mecanism explaining the probable link between infection and anemia in African children is haemolysis in the manuscript.

We thank the reviewer for their helpful comments, which we have addressed point-by-point below:

Major Comment:

Point 1: In the paper of kassebaum et al published in Blood (2014 123:615-624), "a systematic analysis of global anemia burden from 1990 to 2010", the main cause of anemia in Africa is iron deficiency (prevalence around 20 000/100 000, figure 3) were no mecanism of haemolysis is observed.The second is malaria and the third is shistosomiasis. Anemia linked to Sickle cell disease or Thalassemia represent less than 5000/100 000  in 2010. Regarding this figure, more than 1/3 of the anemia is linked to iron deficiency where no Haemolysis is observed.s

Response 1: Thank you. We agree that iron deficiency is a leading cause of anaemia globally and in sub-Saharan Africa. However, iron deficiency has been less commonly observed in severely anaemic children [1-3]. Some epidemiological studies have reported no association between iron deficiency and severe anaemia [2], while others have found that iron deficient individuals are less likely to be severely anaemic [1, 4]. Nevertheless, it is possible that severe iron deficiency anaemia may contribute to the burden of severe anaemia in African children and we have now included a paragraph more specifically addressing severe anaemia aetiologies and have also expanded our discussion on how mechanisms other than haemolysis might contribute to risk of bacterial infection. We have made the following changes to the manuscript:

Introduction, page 2, lines 48-50: “Iron deficiency is a common cause of anaemia in sub-Saharan Africa [5], but is less frequently observed in children with severe anaemia [1, 2, 6].”

The severe anaemia – bacteraemia hypothesis, page 3, lines 126-130: “Although iron deficiency is common in sub-Saharan Africa [5], epidemiological studies have found either no association between iron deficiency and severe anaemia [2], or that iron deficient children living in areas of very high infectious burden are less likely to be severely anaemic [1, 4]. Infections and haemoglobinopathies are strongly associated with severe anaemia [1, 3], and malaria may be the commonest cause of paediatric severe anaemia in sub-Saharan Africa [3, 7-9].

Point 2: The author should add a specific paragraph in the introduction to disccuss the mecanism of infection when the iron and ferritine levels are low and not available for the pathogenic bacteria.

Response 2: Thank you for the suggestion. We have now added a specific paragraph to discuss possible mechanisms of infection when iron and ferritin levels are low as follows:

The severe anaemia – bacteraemia hypothesis, page 3, lines 131-144:In African children, iron deficiency has been reported to protect against infections such as malaria [10, 11]. Unlike other aetiologies of severe anaemia [12-16], there is limited data on the risk of invasive bacterial infections in children with absolute iron deficiency. While iron deficiency may restrict bacterial growth due to limited iron availability [17], very low iron levels may also negatively impact on immune responses to invasive bacterial infections [18, 19]. Iron is required for the development and effector functions of immune cells, including proliferation of T cells and formation of ROS through the Fenton reaction [18, 20, 21]. Severe anaemia and iron deficiency also strongly downregulate the antimicrobial hormone hepcidin even during infection thus facilitating release of iron from storage cells and subsequent loss of hepcidin-induced ‘hypoferraemia of infection’ and ‘nutritional immunity’ [22, 23]. In a population-based survey of Norwegian adults, iron deficiency was associated with increased risk of bloodstream infections [24], possibly through immune dysregulation [25]. The study did not report some measurements including haemoglobin, hepcidin, and ferritin levels, which limits how the data may be interpreted in the context of severe anaemia.

Point 3: Moreover, in the inflamatory process, Haemolysis is not the main mecanism leading to anemia.

Response 3: Thank you. We agree that there are a number of mechanisms that might lead to severe anaemia in addition to haemolysis. One such mechanism is chronic inflammation in which we might observe persistently raised hepcidin levels and increased ferritin levels.

Introduction, page 2, lines 54-59: “Anaemia of inflammation (low serum iron and normal/high ferritin levels) is found in patients with infections (parasitic, bacterial, viral, and fungal), cancer, or autoimmune disorders [26, 27], and is thought to be induced by the hepatic iron regulatory hormone hepcidin [28]. While persistently raised hepcidin levels may protect against invading pathogens [29, 30], enhanced iron sequestration may increase susceptibility to intracellular infections [31].”

The severe anaemia – bacteraemia hypothesis, page 4, lines 144-155: “On the other hand, chronic infection may induce functional iron deficiency. Such “hypoferraemia of infection” is a host defence strategy to limit iron availability from invading pathogens [32], and involves sustained production of proinflammatory cytokines such as interferon-gamma (IFN-γ), tumour necrosis factor-alpha (TNF-α), interleukin (IL)-1 and IL-6 [27]. These proinflammatory cytokines can exacerbate existing severe anaemia by promoting dyserythropoiesis, iron sequestration and erythrophagocytosis [27, 33]. IL-6 promotes the production of hepcidin [28], and has been reported to be upregulated in children with severe malarial anaemia [34, 35]. However, ERFE may have a stronger negative effect on hepcidin production during severe anaemia [36], and low hepcidin levels have been observed in anaemic children and young women with concomitant inflammation [23, 37]. Whether and how IL-6 or other proinflammatory cytokines may increase iron sequestration independently of hepcidin in severely anaemic children remain unknown."

Point 4: The authors should focus on the hemolytic anemia only, like in sickle cell, thallassemia or malaria for example..

Response 4: Thank you for your suggestion. Although severe anaemia due to haemolysis is common in African children [1-4, 7-9] and has strong associations with bacterial infection [12-15], studies suggest that other causes of severe anaemia may also lead to increased bacterial infections. Thus, we also discuss other pathways that might lead to bacterial infection in children with non-haemolytic anaemias including immune dysregulation and gut permeability. Further research is needed to test the hypothesis that severe anaemia leads to bacterial infection in different aetiologies of severe anaemia. We have made the following changes to the manuscript:

The severe anaemia – bacteraemia hypothesis, page 4, lines 166-176: “In model studies, neonatal mice with severe anaemia had a persistent increase in intestinal permeability and electron micrographs showed abnormalities of epithelial adherens junctions probably due to destabilization of the E-cadherin mRNA [38] or decreased expression of the tight junction protein zonula occludens-1 (ZO-1) [39]. Phlebotomy-induced severe anaemia was associated with increased intestinal mucosal hypoxia and production of IFN-γ by intestinal macrophages, which may contribute to increased gut permeability and development of necrotizing enterocolitis [39]. Depletion of intestinal macrophages ablated the effects of severe anaemia on intestinal barrier activity. Whether other immune cells, including neutrophils [40], also contribute to such necrotizing enterocolitis remains unknown. Gut inflammation may also directly induce dysbiosis, through the actions of ROS, calprotectin, and other inflammatory mediators [41-43].

The severe anaemia – bacteraemia hypothesis, page 4, lines 187-195: “Protection against invasive bacterial infection relies on a coordinated and regulated innate and adaptive immune response. In the initial stages of infection, local macrophages engulf and destroy the invading pathogens, and produce cytokines and chemokines to induce an inflammatory response. Monocytes and neutrophils are rapidly recruited to the site of infection. These cells destroy phagocytosed organisms effectively using the NADPH oxidase-dependent ROS production [44, 45]. Neutrophils can also destroy extracellular organisms through degranulation, releasing the neutrophil-extracellular traps (NETs), or production of ROS [44, 46]. Severe anaemia due to iron deficiency may promote impaired development and apoptosis of immune cells [18, 20, 21], including neutrophil hypersegmentation [47] and impaired neutrophil/monocyte oxidative burst [48, 49].”

Minor comments:

Point 1: -In the third paragraph (ligne 269), the subpart should be 3.1 and not "a)" etc..

Response 1: We have amended section 3 subheadings as appropriate to use the 3.1, 3.2 and 3.3 format.

Point 2: - the author should precised that rs334 is a mutation in the beta globin gene.

Response 2: We have included the description of “beta globin gene” in parenthesis after rs334.

Mendelian randomisation, page 10, lines 440-442: “One such genetic variant is rs334 (beta globin gene) in which homozygotes (HbSS or sickle cell disease) have an increased risk of S. pneumoniae, NTS, and H. influenzae.”

Point 3: - Another hypothesis should be take into account. Indeed, the  leucocyte count, and more specifically the neutrophils count should be of interest. For example, not only hemoglobin level but a CBC + DIFF  overview is mandatory to explain infection. The neutrophils are key cells to struggle with pathogenic bacteria.

Response 3: Thank you for the suggestion. We have added a discussion on how severe anaemia may influence neutrophil development, counts, recruitment, and influx into the site of infection.

The severe anaemia – bacteraemia hypothesis, page 4, lines 195-205: “There is limited clinical research regarding the effect of severe anaemia on white blood cell differential count. In some case reports, iron deficiency anaemia has been associated with neutropenia [50, 51]. The effects on neutrophils are supported by murine models of severe malaria anaemia, which reported reduced neutrophil influx and lower proinflammatory cytokine (IL-17 and IFN-?) levels [52, 53], and higher levels of the anti-inflammatory cytokine IL-10 (IL-10) [54, 55]. Elevated IL-6 levels have been observed in field studies of children with severe malarial anaemia [34, 35] and IL-6 may be upregulated when hepcidin levels are low [56] further inhibiting neutrophil influx [57]. IL-6 is an important checkpoint regulator of neutrophil trafficking and promotes clearance of neutrophils and recruitment of monocytes [57, 58]. Downregulation of neutrophil responses may contribute to poor clearance of invasive bacterial pathogens in children with severe anaemia.”

References

  1. Calis, J.C., et al., Severe anemia in Malawian children. N Engl J Med, 2008. 358(9): p. 888-99. DOI: 10.1056/NEJMoa072727.
  2. Foote, E.M., et al., Determinants of anemia among preschool children in rural, western Kenya. Am J Trop Med Hyg, 2013. 88(4): p. 757-64. DOI: 10.4269/ajtmh.12-0560.
  3. Simbauranga, R.H., et al., Prevalence and factors associated with severe anaemia amongst under-five children hospitalized at Bugando Medical Centre, Mwanza, Tanzania. BMC Hematol, 2015. 15: p. 13. DOI: 10.1186/s12878-015-0033-5.
  4. Engle-Stone, R., et al., Predictors of anemia in preschool children: Biomarkers Reflecting Inflammation and Nutritional Determinants of Anemia (BRINDA) project. Am J Clin Nutr, 2017. 106(Suppl 1): p. 402s-415s. DOI: 10.3945/ajcn.116.142323.
  5. Kassebaum, N.J., The Global Burden of Anemia. Hematol Oncol Clin North Am, 2016. 30(2): p. 247-308. DOI: 10.1016/j.hoc.2015.11.002.
  6. van Hensbroek, M.B., F. Jonker, and I. Bates, Severe acquired anaemia in Africa: new concepts. Br J Haematol, 2011. 154(6): p. 690-5. DOI: 10.1111/j.1365-2141.2011.08761.x.
  7. Newton, C.R., et al., Severe anaemia in children living in a malaria endemic area of Kenya. Trop Med Int Health, 1997. 2(2): p. 165-78.
  8. Adegoke, S., et al., Determinants of mortality in Nigerian children with severe anaemia. S Afr Med J, 2012. 102(10): p. 807-10. DOI: 10.7196/samj.5240.
  9. Muoneke, V. and R. ChidiIbekwe, Prevalence and aetiology of severe anaemia in under-5 children in Abakaliki South Eastern Nigeria. Pediatr Therapeut, 2011. 1(3): p. 107.
  10. Nyakeriga, A.M., et al., Iron deficiency and malaria among children living on the coast of Kenya. J Infect Dis, 2004. 190(3): p. 439-47. DOI: 10.1086/422331.
  11. Muriuki, J.M., et al., Iron Status and Associated Malaria Risk Among African Children. Clin Infect Dis, 2019. 68(11): p. 1807-1814. DOI: 10.1093/cid/ciy791.
  12. Bronzan, R.N., et al., Bacteremia in Malawian children with severe malaria: prevalence, etiology, HIV coinfection, and outcome. J Infect Dis, 2007. 195(6): p. 895-904. DOI: 10.1086/511437.
  13. Graham, S.M., et al., The pattern of bacteraemia in children with severe malaria. Malawi Med J, 2002. 14(1): p. 11-5.
  14. Church, J. and K. Maitland, Invasive bacterial co-infection in African children with Plasmodium falciparum malaria: a systematic review. BMC Med, 2014. 12(1): p. 31. DOI: 10.1186/1741-7015-12-31.
  15. Williams, T.N., et al., Bacteraemia in Kenyan children with sickle-cell anaemia: a retrospective cohort and case-control study. Lancet, 2009. 374(9698): p. 1364-70. DOI: 10.1016/s0140-6736(09)61374-x.
  16. Ramakrishnan, M., et al., Increased risk of invasive bacterial infections in African people with sickle-cell disease: a systematic review and meta-analysis. Lancet Infect Dis, 2010. 10(5): p. 329-37. DOI: 10.1016/s1473-3099(10)70055-4.
  17. Michels, K.R., et al., The Role of Iron in the Susceptibility of Neonatal Mice to Escherichia coli K1 Sepsis. J Infect Dis, 2019. 220(7): p. 1219-1229. DOI: 10.1093/infdis/jiz282.
  18. Hassan, T.H., et al., Impact of iron deficiency anemia on the function of the immune system in children. Medicine (Baltimore), 2016. 95(47): p. e5395. DOI: 10.1097/md.0000000000005395.
  19. Darshan, D., et al., Severe iron deficiency blunts the response of the iron regulatory gene Hamp and pro-inflammatory cytokines to lipopolysaccharide. Haematologica, 2010. 95(10): p. 1660-7. DOI: 10.3324/haematol.2010.022426.
  20. Jiang, Y., et al., Iron-dependent histone 3 lysine 9 demethylation controls B cell proliferation and humoral immune responses. Nat Commun, 2019. 10(1): p. 2935. DOI: 10.1038/s41467-019-11002-5.
  21. Das, I., et al., Impact of iron deficiency anemia on cell-mediated and humoral immunity in children: A case control study. J Nat Sci Biol Med, 2014. 5(1): p. 158-63. DOI: 10.4103/0976-9668.127317.
  22. Ganz, T. and E. Nemeth, Hepcidin and iron homeostasis. Biochim Biophys Acta, 2012. 1823(9): p. 1434-43. DOI: 10.1016/j.bbamcr.2012.01.014.
  23. Jonker, F.A., et al., Low hepcidin levels in severely anemic malawian children with high incidence of infectious diseases and bone marrow iron deficiency. PLoS One, 2013. 8(12): p. e78964. DOI: 10.1371/journal.pone.0078964.
  24. Mohus, R.M., et al., Association of iron status with the risk of bloodstream infections: results from the prospective population-based HUNT Study in Norway. Intensive Care Med, 2018. 44(8): p. 1276-1283. DOI: 10.1007/s00134-018-5320-8.
  25. Swenson, E.R., R. Porcher, and M. Piagnerelli, Iron deficiency and infection: another pathway to explore in critically ill patients? Intensive Care Med, 2018. 44(12): p. 2260-2262. DOI: 10.1007/s00134-018-5438-8.
  26. Lambert, J. and P. Beris, Pathophysiology and differential diagnosis of anaemia. Beaumont C, Béris P, Beuzard Y, Brugnara C. The Handbook on disorders of erythropoiesis, erythrocytes and iron metabolism. Paris: European school of haematology, 2009: p. 108-41.
  27. Nemeth, E. and T. Ganz, Anemia of inflammation. Hematol Oncol Clin North Am, 2014. 28(4): p. 671-81, vi. DOI: 10.1016/j.hoc.2014.04.005.
  28. Nemeth, E., et al., IL-6 mediates hypoferremia of inflammation by inducing the synthesis of the iron regulatory hormone hepcidin. Journal of Clinical Investigation, 2004. 113(9): p. 1271-1276. DOI: 10.1172/jci200420945.
  29. Arezes, J., et al., Hepcidin-induced hypoferremia is a critical host defense mechanism against the siderophilic bacterium Vibrio vulnificus. Cell Host Microbe, 2015. 17(1): p. 47-57. DOI: 10.1016/j.chom.2014.12.001.
  30. Wang, H.Z., et al., Hepcidin is regulated during blood-stage malaria and plays a protective role in malaria infection. The Journal of immunology, 2011. 187(12): p. 6410-6. DOI: 10.4049/jimmunol.1101436.
  31. Lokken, K.L., et al., Malaria Parasite-Mediated Alteration of Macrophage Function and Increased Iron Availability Predispose to Disseminated Nontyphoidal Salmonella Infection. Infect Immun, 2018. 86(9): p. e00301-18. DOI: 10.1128/iai.00301-18.
  32. Nairz, M., et al., Iron and innate antimicrobial immunity-Depriving the pathogen, defending the host. J Trace Elem Med Biol, 2018. 48: p. 118-133. DOI: 10.1016/j.jtemb.2018.03.007.
  33. de Bruin, A.M., C. Voermans, and M.A. Nolte, Impact of interferon-gamma on hematopoiesis. Blood, 2014. 124(16): p. 2479-86. DOI: 10.1182/blood-2014-04-568451.
  34. Mandala, W.L., et al., Cytokine Profiles in Malawian Children Presenting with Uncomplicated Malaria, Severe Malarial Anemia, and Cerebral Malaria. Clin Vaccine Immunol, 2017. 24(4). DOI: 10.1128/cvi.00533-16.
  35. Burte, F., et al., Circulatory hepcidin is associated with the anti-inflammatory response but not with iron or anemic status in childhood malaria. Blood, 2013. 121(15): p. 3016-22. DOI: 10.1182/blood-2012-10-461418.
  36. Latour, C., et al., Erythroferrone contributes to hepcidin repression in a mouse model of malarial anemia. Haematologica, 2017. 102(1): p. 60-68. DOI: 10.3324/haematol.2016.150227.
  37. Stoffel, N.U., et al., The opposing effects of acute inflammation and iron deficiency anemia on serum hepcidin and iron absorption in young women. Haematologica, 2019. 104(6): p. 1143-1149. DOI: 10.3324/haematol.2018.208645.
  38. MohanKumar, K., et al., Severe neonatal anemia increases intestinal permeability by disrupting epithelial adherens junctions. Am J Physiol Gastrointest Liver Physiol, 2020. 318(4): p. G705-g716. DOI: 10.1152/ajpgi.00324.2019.
  39. Arthur, C.M., et al., Anemia induces gut inflammation and injury in an animal model of preterm infants. Transfusion, 2019. 59(4): p. 1233-1245. DOI: 10.1111/trf.15254.
  40. Spees, A.M., et al., Neutrophils are a source of gamma interferon during acute Salmonella enterica serovar Typhimurium colitis. Infect Immun, 2014. 82(4): p. 1692-7. DOI: 10.1128/iai.01508-13.
  41. Winter, S.E., et al., Gut inflammation provides a respiratory electron acceptor for Salmonella. Nature, 2010. 467(7314): p. 426-9. DOI: 10.1038/nature09415.
  42. Liu, J.Z., et al., Zinc sequestration by the neutrophil protein calprotectin enhances Salmonella growth in the inflamed gut. Cell Host Microbe, 2012. 11(3): p. 227-39. DOI: 10.1016/j.chom.2012.01.017.
  43. Goethel, A., K. Croitoru, and D.J. Philpott, The interplay between microbes and the immune response in inflammatory bowel disease. J Physiol, 2018. 596(17): p. 3869-3882. DOI: 10.1113/jp275396.
  44. Kruger, P., et al., Neutrophils: Between host defence, immune modulation, and tissue injury. PLoS Pathog, 2015. 11(3): p. e1004651. DOI: 10.1371/journal.ppat.1004651.
  45. Gogoi, M., M.M. Shreenivas, and D. Chakravortty, Hoodwinking the Big-Eater to Prosper: The Salmonella -Macrophage Paradigm. Journal of Innate Immunity, 2019. 11(3): p. 289-299. DOI: http://dx.doi.org/10.1159/000490953.
  46. Brinkmann, V., et al., Neutrophil extracellular traps kill bacteria. Science, 2004. 303(5663): p. 1532-5. DOI: 10.1126/science.1092385.
  47. Westerman, D.A., D. Evans, and J. Metz, Neutrophil hypersegmentation in iron deficiency anaemia: a case-control study. Br J Haematol, 1999. 107(3): p. 512-5. DOI: 10.1046/j.1365-2141.1999.01756.x.
  48. Berrak, S.G., et al., The effects of iron deficiency on neutrophil/monocyte apoptosis in children. Cell Prolif, 2007. 40(5): p. 741-54. DOI: 10.1111/j.1365-2184.2007.00460.x.
  49. Murakawa, H., et al., Iron deficiency and neutrophil function: different rates of correction of the depressions in oxidative burst and myeloperoxidase activity after iron treatment. Blood, 1987. 69(5): p. 1464-8.
  50. Abdelmahmuod, E., M.A. Yassin, and M. Ahmed, Iron Deficiency Anemia-Induced Neutropenia in Adult Female. Cureus, 2020. 12(6): p. e8899. DOI: 10.7759/cureus.8899.
  51. Abuirmeileh, A., A. Bahnassi, and A. Abuirmeileh, Unexplained chronic leukopenia treated with oral iron supplements. Int J Clin Pharm, 2014. 36(2): p. 264-7. DOI: 10.1007/s11096-013-9897-2.
  52. Mooney, J.P., et al., The mucosal inflammatory response to non-typhoidal Salmonella in the intestine is blunted by IL-10 during concurrent malaria parasite infection. Mucosal Immunol, 2014. 7(6): p. 1302-11. DOI: 10.1038/mi.2014.18.
  53. Mooney, J.P., L.J. Galloway, and E.M. Riley, Malaria, anemia, and invasive bacterial disease: A neutrophil problem? J Leukoc Biol, 2019. 105(4): p. 645-655. DOI: 10.1002/jlb.3ri1018-400r.
  54. Lokken, K.L., et al., Malaria parasite infection compromises control of concurrent systemic non-typhoidal Salmonella infection via IL-10-mediated alteration of myeloid cell function. PLoS Pathog, 2014. 10(5): p. e1004049. DOI: 10.1371/journal.ppat.1004049.
  55. Roux, C.M., et al., Both hemolytic anemia and malaria parasite-specific factors increase susceptibility to Nontyphoidal Salmonella enterica serovar typhimurium infection in mice. Infect Immun, 2010. 78(4): p. 1520-7. DOI: 10.1128/iai.00887-09.
  56. Pagani, A., et al., Low hepcidin accounts for the proinflammatory status associated with iron deficiency. Blood, 2011. 118(3): p. 736-46. DOI: 10.1182/blood-2011-02-337212.
  57. Kaplanski, G., et al., IL-6: a regulator of the transition from neutrophil to monocyte recruitment during inflammation. Trends Immunol, 2003. 24(1): p. 25-9. DOI: 10.1016/s1471-4906(02)00013-3.
  58. Fielding, C.A., et al., IL-6 regulates neutrophil trafficking during acute inflammation via STAT3. J Immunol, 2008. 181(3): p. 2189-95. DOI: 10.4049/jimmunol.181.3.2189.

Reviewer 2 Report

the work of Abuga and colleagues is very interesting. It comprehensively describes the relationship between severe anemia and bacterial infections. I have only a few minor observations, written later, in order to deepen the link with clinical features and pathogenesis of the bacterial infections analysed into the review.
Moreover, the major hypothesis of the article linked, in an interesting way, bacterial infection to severe haemolytic anaemia. Haemolytic anaemia is one of the pathogenic mechanisms that underline anaemia in African children. Is the hypothesis applicable also to other kinds of anaemia (e.g. hyporegenerative anaemia as in the iron or B12 deficit?)? if not, probably the title may be changed, for example. To “ How severe haemolytic anaemia might influence the risk of  invasive bacterial infections in African children”

Page 1 line 7 remove the word (pneumococci)

Page 2 line 78 authors may better explain how the alteration in the sequestration of iron into the macrophages may promote the dissemination of a bacterial infection

Page 3 line 84 the high risk of NTS and E.coli in anaemic children is real regardless of the cause of the anaemia (infection, haemoglobinopathies, nutritional deficit…) or is linked to a specific kind of anaemia?

Page 3 line 89 the hypothesis is very interesting, but not all kind of anaemia in African children are haemolytic anaemia: how to explain the link between anaemia and intestinal increased permeability in non-haemolytic anaemia, in a setting of low iron concentrations especially in gut cels?

Page 3 line 106 the relation between HO-1, anaemia and immune dysregulation in unclear. The authors may revise this part, to better explain how severe anaemia modulates HO-1 and immune responses

Page 5 line 161 which kind of NTS infection are related to severe anaemia? Sepsis? Gastrointestinal infection? Are there any studies in paediatric age about this?

Page 5 line 179: E.coli infection is strictly related to iron overload, as in haemolytic anaemia or in the settings of strong (intramuscular or intravenous) iron supplementation. Is E.coli infection more present also in hyporegenerative anaemia?

Furthermore, the paragraph is dedicated to which kind of infection? Sepsis? And what about urinary tract infection and pyelonephritis, that are the most common and most important e.coli infection in paediatric age?

Page 6 line 235: as for others pathogens, authors may provide data, is available, about the different kind of infections caused by staph. Aureus in children. For example, skin and soft tissue infection are very different from bloodstream infections from a pathogenic point of view. So, are there any studies about a different kind of Stah. Aureus infections in children and its relationship with severe or haemolytic anaemia?

Author Response

The work of Abuga and colleagues is very interesting. It comprehensively describes the relationship between severe anemia and bacterial infections. I have only a few minor observations, written later, in order to deepen the link with clinical features and pathogenesis of the bacterial infections analysed into the review. 

We thank the reviewer for their helpful comments, which we have addressed point-by-point below:

Point 1: Moreover, the major hypothesis of the article linked, in an interesting way, bacterial infection to severe haemolytic anaemia. Haemolytic anaemia is one of the pathogenic mechanisms that underline anaemia in African children. Is the hypothesis applicable also to other kinds of anaemia (e.g. hyporegenerative anaemia as in the iron or B12 deficit?)? if not, probably the title may be changed, for example. To “ How severe haemolytic anaemia might influence the risk of  invasive bacterial infections in African children”

Response 1: Thank you for the suggestion. Although severe anaemia due to haemolysis is common in African children [1-7] and has strong associations with bacterial infection [8-11], studies suggest that other causes of severe anaemia may also lead to increased bacterial infections. Thus, we also discuss other pathways that might lead to bacterial infection in children with non-haemolytic anaemias including immune dysregulation and gut permeability. We have added the following paragraph on how severe anaemia due to iron deficiency may increase risk of invasive bacterial infections.

The severe anaemia – bacteraemia hypothesis, page 3, lines 131-144:  “In African children, iron deficiency has been reported to protect against infections such as malaria [12, 13]. Unlike other aetiologies of severe anaemia [8-11, 14], there is limited data on the risk of invasive bacterial infections in children with absolute iron deficiency. While iron deficiency may restrict bacterial growth due to limited iron availability [15], very low iron levels may also negatively impact on immune responses to invasive bacterial infections [16, 17]. Iron is required for the development and effector functions of immune cells, including proliferation of T cells and formation of ROS through the Fenton reaction [16, 18, 19]. Severe anaemia and iron deficiency also strongly downregulate the antimicrobial hormone hepcidin even during infection thus facilitating release of iron from storage cells and subsequent loss of hepcidin-induced ‘hypoferraemia of infection’ and ‘nutritional immunity’ [20, 21]. In a population-based survey of Norwegian adults, iron deficiency was associated with increased risk of bloodstream infections [22], possibly through immune dysregulation [23]. The study did not report some measurements including haemoglobin, hepcidin, and ferritin levels, which limits how the data may be interpreted in the context of severe anaemia.

Point 2: Page 1 line 7 remove the word (pneumococci)

Response 2: We have now removed the word pneumococci as suggested.

Introduction, page 1, lines 36-38: “The commonest bacterial isolates observed in African children are Streptococcus pneumoniae, Staphylococcus aureus, non-typhoidal Salmonellae (NTS), Haemophilus influenzae and Escherichia coli.”

Point 3: Page 2 line 78 authors may better explain how the alteration in the sequestration of iron into the macrophages may promote the dissemination of a bacterial infection

Response 3: We have now expanded our explanation of how alteration in sequestration of iron into macrophages may promote bacterial infection. Iron sequestration into macrophages is a defence strategy aimed at protecting against extracellular pathogens [24], but may promote infections with intracellular pathogens [25, 26]. We have edited the section to improve on clarity as follows:

The severe anaemia – bacteraemia hypothesis, page 3, lines 101-122: “In circulation, the net iron concentration is maintained through efficient macrophage-recycling of iron from senescent or damaged red blood cells, and the effective use of iron in the bone marrow for erythropoiesis. Most intracellular iron is complexed to haem or the iron-storage protein ferritin, while extracellular iron is bound to high-affinity chaperone proteins including transferrin, haptoglobin, hemopexin, lipocalin-2 and lactoferrin. The hepatic hormone hepcidin is the master iron regulator, and maintains iron homeostasis by controlling absorption of dietary iron, release of iron from storage cells, and sequestration of recycled iron in macrophages [27]. Infections by extracellular pathogens result in cellular iron import via various receptors including those of transferrin, lipocalin-2, haem-haemopexin (CD91), and haemoglobin-haptoglobin (CD163) complexes. Elevated hepcidin levels further ensure iron is maintained intracellularly by degrading the sole iron exporter, ferroportin [28]. Reduced availability of iron in plasma “starves” invading pathogens and protects against extracellular infection. Low hepcidin levels, observed during severe anaemia [21, 29-31], may undermine this nutritional immunity. Intracellular infections, on the other hand, are associated with reduction in cellular haem-iron content and rely on suppression of iron import into macrophages and/or increased iron export out of cells. An additional defence strategy involves iron export out of phagolysosomes, such as the Salmonella-containing vacuole, using the natural resistance-associated macrophage protein 1 (Nramp1). Concomitant infections that promote iron sequestration into macrophages may disrupt these iron regulation strategies [25, 26] and predispose African children with severe anaemia to an increased risk of intracellular bacterial infections.”

Point 4: Page 3 line 84 the high risk of NTS and E. coli in anaemic children is real regardless of the cause of the anaemia (infection, haemoglobinopathies, nutritional deficit…) or is linked to a specific kind of anaemia?

Response 4: Thank you for the observation. In African children, the association between severe anaemia and enteric organisms has been observed in severely anaemic children generally [1], as well as in those with various severe anaemia aetiologies including sickle cell disease [9], malaria [10, 32], HIV [10, 33], and malnutrition [34, 35]. Further research is needed into this area. We have added the second part of the sentence below:

The severe anaemia – bacteraemia hypothesis, page 4, line 158-161: “However, African children with severe anaemia have a high risk of bacteraemia due to enteric organisms, particularly NTS and E. coli [36-39], and this has been observed in severely anaemic children generally [1] and in those with underlying sickle cell disease [9], malaria [10, 32], HIV [10, 33], and malnutrition [34, 35].”

Point 5: Page 3 line 89 the hypothesis is very interesting, but not all kind of anaemia in African children are haemolytic anaemia:: how to explain the link between anaemia and intestinal increased permeability in non-haemolytic anaemia, in a setting of low iron concentrations especially in gut cels?

Response 5: Thank you for the comment. We have now included a discussion on how sustained inflammatory responses during severe anaemia may increase intestinal permeability and/or necrotizing enterocolitis as follows:

The severe anaemia – bacteraemia hypothesis, page 4, lines 166-176: “In model studies, neonatal mice with severe anaemia had a persistent increase in intestinal permeability and electron micrographs showed abnormalities of epithelial adherens junctions probably due to destabilization of the E-cadherin mRNA [40] or decreased expression of the tight junction protein zonula occludens-1 (ZO-1) [41]. Phlebotomy-induced severe anaemia was associated with increased intestinal mucosal hypoxia and production of IFN-γ by intestinal macrophages, which may contribute to increased gut permeability and development of necrotizing enterocolitis [41]. Depletion of intestinal macrophages ablated the effects of severe anaemia on intestinal barrier activity. Whether other immune cells, including neutrophils [42], also contribute to such necrotizing enterocolitis remains unknown. Gut inflammation may also directly induce dysbiosis, through the actions of ROS, calprotectin, and other inflammatory mediators [43-45].”

Point 6: Page 3 line 106 the relation between HO-1, anaemia and immune dysregulation in unclear. The authors may revise this part, to better explain how severe anaemia modulates HO-1 and immune responses.

Response 6: Thank you for the comment. We have now given a clearer explanation of how severe anaemia modulates HO-1 and immune responses in our paragraph as below:

The severe anaemia – bacteraemia hypothesis, page 5, lines 214-228: “The haem-catabolizing enzyme, haem oxygenase-1 (HO-1), may also downregulate immune responses to bacterial infections. HO-1 is normally expressed at low levels in most tissues but is highly induced by inflammation, hypoxia and other stimuli. In conditions with increased haemolysis, HO-1 is induced to break down the elevated free haem into equimolar amounts of carbon monoxide, biliverdin, and ferrous iron. Whilst its induction reduces oxidative damage by free haem, HO-1 is associated with reduced elimination of pathogens including systemic NTS, malaria and leishmaniasis [46-48]. This may be a result of the direct tolerogenic effects of HO-1 on the immune system [49], or due to the actions of its products. Biliverdin and carbon monoxide are anti-inflammatory and scavenge radical molecules that kill intracellular bacteria [47, 50]. Moreover, intracellular iron inhibits the activity of IFN-γ in a dose-dependent manner [51, 52]. IFN-γ is central to the control of intracellular pathogens by inducing ROS generation through the nitric oxide synthase pathway [53]. Inhibiting the expression of IFN-γ increases availability of iron for intracellular pathogens by increasing uptake of transferrin-bound iron into macrophages [54] and storage of iron in ferritin [55]. Iron may also inhibit the expression of other inflammatory mediators including tumour-necrosis factor and nitric oxide synthase [56, 57].”

Point 7: Page 5 line 161 which kind of NTS infection are related to severe anaemia? Sepsis? Gastrointestinal infection? Are there any studies in paediatric age about this?

Response 7: Thank you for the comment. Most studies have found an association between severe anaemia and NTS septicaemia in African children [36-38, 58], and that young age is a risk factor for mortality in children with NTS bacteraemia [35, 36]. There is limited information on the relationship between severe anaemia and NTS gastrointestinal infection in African children, and an early study did not find an association between faecal NTS and malaria seasonality [58]. We have added the following statement to our manuscript:

Non-typhoidal Salmonellae, page 7, line 277: “NTS bacteraemia is strongly associated with severe anaemia in African children [36-38].”

Point 8: Page 5 line 179: E.coli infection is strictly related to iron overload, as in haemolytic anaemia or in the settings of strong (intramuscular or intravenous) iron supplementation. Is E.coli infection more present also in hyporegenerative anaemia?

Response 8: Thank you for the comment. There is limited information on E. coli infections in children with hyporegenerative anaemias. Aplastic anaemia is associated with increased E. coli infections [59]. In model studies, iron deficiency is associated with protection against E. coli sepsis [15, 60], however severe iron deficiency anaemia reduces hepcidin production even during inflammation [21, 61], and very low hepcidin levels are associated with increased susceptibility to E. coli infections [60, 62]. Iron deficiency may also be associated with poor immune development and responses to infections [16, 18, 19], including E. coli. Further work is needed to determine if these findings can be translated to human populations. We have added the following statements in our manuscript:

Escherichia coli, page 7, lines 304-311: “European children given intramuscular iron dextran [63], patients with aplastic anaemia [59], and an adult with iron overload [64] had increased susceptibility to E. coli sepsis. Consequently, it is plausible that increased serum and tissue iron levels during haemolysis and severe anaemia may similarly increase risk of E. coli infections. In model studies, iron deficiency is associated with protection against E. coli sepsis [15, 60], however severe iron deficiency anaemia reduces hepcidin production even during inflammation [21, 61], and very low hepcidin levels have been associated with increased susceptibility to E. coli infections [60, 62]. Iron deficiency may also be associated with poor immune development and responses to infections [16, 18, 19].

Point 9: Furthermore, the paragraph is dedicated to which kind of infection? Sepsis? And what about urinary tract infection and pyelonephritis, that are the most common and most important e.coli infection in paediatric age?

Response 9: Thank you for the suggestion. We restricted this hypothesis to invasive (bloodstream) infections based on the observed strong associations between severe anaemia and bacteraemia in African children. Discussion on urinary tract infections (UTIs) and pyelonephritis were beyond the scope of this hypothesis. We believe that the mechanisms discussed may also apply to E. coli causing UTIs to some extent, although the urinary tract environment and E. coli pathogens colonizing it may have different features from the bloodstream infections [65, 66]. To clarify the scope of this hypothesis, we have edited the following statement:

The severe anaemia – bacteraemia hypothesis, page 6, lines 260-262: “Below we discuss iron acquisition strategies of the pathogenic bacteria that are commonly isolated in blood cultures of African children [67-69], and how severe anaemia might influence iron availability for these organisms.”

Point 10: Page 6 line 235: as for others pathogens, authors may provide data, is available, about the different kind of infections caused by staph. Aureus in children. For example, skin and soft tissue infection are very different from bloodstream infections from a pathogenic point of view. So, are there any studies about a different kind of Stah. Aureus infections in children and its relationship with severe or haemolytic anaemia?

Response 10: Thank you for the comment. We restricted this hypothesis to bloodstream infections based on the observed strong associations between severe anaemia and bacteraemia in African children. Discussion on skin and soft tissue infections were beyond the scope of this hypothesis. The iron acquisition dynamics for skin and soft tissue S. aureus infection may be different from blood stream infections, although there is limited information on the precise strategies or association with severe anaemia in African children. In-vitro studies suggest that S. aureus can utilise myoglobin [70], and this may be an important iron source during skin and soft tissue infections [71].

References

  1. Calis, J.C., et al., Severe anemia in Malawian children. N Engl J Med, 2008. 358(9): p. 888-99. DOI: 10.1056/NEJMoa072727.
  2. Foote, E.M., et al., Determinants of anemia among preschool children in rural, western Kenya. Am J Trop Med Hyg, 2013. 88(4): p. 757-64. DOI: 10.4269/ajtmh.12-0560.
  3. Simbauranga, R.H., et al., Prevalence and factors associated with severe anaemia amongst under-five children hospitalized at Bugando Medical Centre, Mwanza, Tanzania. BMC Hematol, 2015. 15: p. 13. DOI: 10.1186/s12878-015-0033-5.
  4. Newton, C.R., et al., Severe anaemia in children living in a malaria endemic area of Kenya. Trop Med Int Health, 1997. 2(2): p. 165-78.
  5. Adegoke, S., et al., Determinants of mortality in Nigerian children with severe anaemia. S Afr Med J, 2012. 102(10): p. 807-10. DOI: 10.7196/samj.5240.
  6. Muoneke, V. and R. ChidiIbekwe, Prevalence and aetiology of severe anaemia in under-5 children in Abakaliki South Eastern Nigeria. Pediatr Therapeut, 2011. 1(3): p. 107.
  7. Engle-Stone, R., et al., Predictors of anemia in preschool children: Biomarkers Reflecting Inflammation and Nutritional Determinants of Anemia (BRINDA) project. Am J Clin Nutr, 2017. 106(Suppl 1): p. 402s-415s. DOI: 10.3945/ajcn.116.142323.
  8. Church, J. and K. Maitland, Invasive bacterial co-infection in African children with Plasmodium falciparum malaria: a systematic review. BMC Med, 2014. 12(1): p. 31. DOI: 10.1186/1741-7015-12-31.
  9. Williams, T.N., et al., Bacteraemia in Kenyan children with sickle-cell anaemia: a retrospective cohort and case-control study. Lancet, 2009. 374(9698): p. 1364-70. DOI: 10.1016/s0140-6736(09)61374-x.
  10. Bronzan, R.N., et al., Bacteremia in Malawian children with severe malaria: prevalence, etiology, HIV coinfection, and outcome. J Infect Dis, 2007. 195(6): p. 895-904. DOI: 10.1086/511437.
  11. Graham, S.M., et al., The pattern of bacteraemia in children with severe malaria. Malawi Med J, 2002. 14(1): p. 11-5.
  12. Nyakeriga, A.M., et al., Iron deficiency and malaria among children living on the coast of Kenya. J Infect Dis, 2004. 190(3): p. 439-47. DOI: 10.1086/422331.
  13. Muriuki, J.M., et al., Iron Status and Associated Malaria Risk Among African Children. Clin Infect Dis, 2019. 68(11): p. 1807-1814. DOI: 10.1093/cid/ciy791.
  14. Ramakrishnan, M., et al., Increased risk of invasive bacterial infections in African people with sickle-cell disease: a systematic review and meta-analysis. Lancet Infect Dis, 2010. 10(5): p. 329-37. DOI: 10.1016/s1473-3099(10)70055-4.
  15. Michels, K.R., et al., The Role of Iron in the Susceptibility of Neonatal Mice to Escherichia coli K1 Sepsis. J Infect Dis, 2019. 220(7): p. 1219-1229. DOI: 10.1093/infdis/jiz282.
  16. Hassan, T.H., et al., Impact of iron deficiency anemia on the function of the immune system in children. Medicine (Baltimore), 2016. 95(47): p. e5395. DOI: 10.1097/md.0000000000005395.
  17. Darshan, D., et al., Severe iron deficiency blunts the response of the iron regulatory gene Hamp and pro-inflammatory cytokines to lipopolysaccharide. Haematologica, 2010. 95(10): p. 1660-7. DOI: 10.3324/haematol.2010.022426.
  18. Jiang, Y., et al., Iron-dependent histone 3 lysine 9 demethylation controls B cell proliferation and humoral immune responses. Nat Commun, 2019. 10(1): p. 2935. DOI: 10.1038/s41467-019-11002-5.
  19. Das, I., et al., Impact of iron deficiency anemia on cell-mediated and humoral immunity in children: A case control study. J Nat Sci Biol Med, 2014. 5(1): p. 158-63. DOI: 10.4103/0976-9668.127317.
  20. Ganz, T. and E. Nemeth, Hepcidin and iron homeostasis. Biochim Biophys Acta, 2012. 1823(9): p. 1434-43. DOI: 10.1016/j.bbamcr.2012.01.014.
  21. Jonker, F.A., et al., Low hepcidin levels in severely anemic malawian children with high incidence of infectious diseases and bone marrow iron deficiency. PLoS One, 2013. 8(12): p. e78964. DOI: 10.1371/journal.pone.0078964.
  22. Mohus, R.M., et al., Association of iron status with the risk of bloodstream infections: results from the prospective population-based HUNT Study in Norway. Intensive Care Med, 2018. 44(8): p. 1276-1283. DOI: 10.1007/s00134-018-5320-8.
  23. Swenson, E.R., R. Porcher, and M. Piagnerelli, Iron deficiency and infection: another pathway to explore in critically ill patients? Intensive Care Med, 2018. 44(12): p. 2260-2262. DOI: 10.1007/s00134-018-5438-8.
  24. Ganz, T., Iron in innate immunity: starve the invaders. Curr Opin Immunol, 2009. 21(1): p. 63-7. DOI: 10.1016/j.coi.2009.01.011.
  25. Kaye, D. and E.W. Hook, The influence of hemolysis on susceptibility to Salmonella infection: Additional observations. J Immunol, 1963. 91: p. 518-27.
  26. Lokken, K.L., et al., Malaria Parasite-Mediated Alteration of Macrophage Function and Increased Iron Availability Predispose to Disseminated Nontyphoidal Salmonella Infection. Infect Immun, 2018. 86(9): p. e00301-18. DOI: 10.1128/iai.00301-18.
  27. Ganz, T. and E. Nemeth, Iron homeostasis in host defence and inflammation. Nat Rev Immunol, 2015. 15(8): p. 500-10. DOI: 10.1038/nri3863.
  28. Nemeth, E., et al., Hepcidin regulates cellular iron efflux by binding to ferroportin and inducing its internalization. Science, 2004. 306(5704): p. 2090-3. DOI: 10.1126/science.1104742.
  29. Kautz, L., et al., Identification of erythroferrone as an erythroid regulator of iron metabolism. Nat Genet, 2014. 46(7): p. 678-84. DOI: 10.1038/ng.2996.
  30. Latour, C., et al., Erythroferrone contributes to hepcidin repression in a mouse model of malarial anemia. Haematologica, 2017. 102(1): p. 60-68. DOI: 10.3324/haematol.2016.150227.
  31. Lee, N., et al., Decreased Hepcidin Levels Are Associated with Low Steady-state Hemoglobin in Children With Sickle Cell Disease in Tanzania. EBioMedicine, 2018. 34: p. 158-164. DOI: 10.1016/j.ebiom.2018.07.024.
  32. Berkley, J.A., et al., HIV infection, malnutrition, and invasive bacterial infection among children with severe malaria. Clin Infect Dis, 2009. 49(3): p. 336-43. DOI: 10.1086/600299.
  33. Lewis, D.K., et al., Treatable factors associated with severe anaemia in adults admitted to medical wards in Blantyre, Malawi, an area of high HIV seroprevalence. Transactions of the Royal Society of Tropical Medicine and Hygiene, 2005. 99(8): p. 561-567.
  34. Page, A.L., et al., Infections in children admitted with complicated severe acute malnutrition in Niger. PLoS One, 2013. 8(7): p. e68699. DOI: 10.1371/journal.pone.0068699.
  35. Mandomando, I., et al., Invasive non-typhoidal Salmonella in Mozambican children. Trop Med Int Health, 2009. 14(12): p. 1467-74. DOI: 10.1111/j.1365-3156.2009.02399.x.
  36. Brent, A.J., et al., Salmonella bacteremia in Kenyan children. Pediatr Infect Dis J, 2006. 25(3): p. 230-6. DOI: 10.1097/01.inf.0000202066.02212.ff.
  37. Graham, S.M., et al., Clinical presentation of non-typhoidal Salmonella bacteraemia in Malawian children. Trans R Soc Trop Med Hyg, 2000. 94(3): p. 310-4. DOI: 10.1016/s0035-9203(00)90337-7.
  38. Biggs, H.M., et al., Invasive Salmonella infections in areas of high and low malaria transmission intensity in Tanzania. Clin Infect Dis, 2014. 58(5): p. 638-47. DOI: 10.1093/cid/cit798.
  39. Sigauque, B., et al., Community-acquired bacteremia among children admitted to a rural hospital in Mozambique. Pediatr Infect Dis J, 2009. 28(2): p. 108-13. DOI: 10.1097/INF.0b013e318187a87d.
  40. MohanKumar, K., et al., Severe neonatal anemia increases intestinal permeability by disrupting epithelial adherens junctions. Am J Physiol Gastrointest Liver Physiol, 2020. 318(4): p. G705-g716. DOI: 10.1152/ajpgi.00324.2019.
  41. Arthur, C.M., et al., Anemia induces gut inflammation and injury in an animal model of preterm infants. Transfusion, 2019. 59(4): p. 1233-1245. DOI: 10.1111/trf.15254.
  42. Spees, A.M., et al., Neutrophils are a source of gamma interferon during acute Salmonella enterica serovar Typhimurium colitis. Infect Immun, 2014. 82(4): p. 1692-7. DOI: 10.1128/iai.01508-13.
  43. Winter, S.E., et al., Gut inflammation provides a respiratory electron acceptor for Salmonella. Nature, 2010. 467(7314): p. 426-9. DOI: 10.1038/nature09415.
  44. Liu, J.Z., et al., Zinc sequestration by the neutrophil protein calprotectin enhances Salmonella growth in the inflamed gut. Cell Host Microbe, 2012. 11(3): p. 227-39. DOI: 10.1016/j.chom.2012.01.017.
  45. Goethel, A., K. Croitoru, and D.J. Philpott, The interplay between microbes and the immune response in inflammatory bowel disease. J Physiol, 2018. 596(17): p. 3869-3882. DOI: 10.1113/jp275396.
  46. Cunnington, A.J., et al., Malaria impairs resistance to Salmonella through heme- and heme oxygenase-dependent dysfunctional granulocyte mobilization. Nat Med, 2011. 18(1): p. 120-7. DOI: 10.1038/nm.2601.
  47. Mitterstiller, A.M., et al., Heme oxygenase 1 controls early innate immune response of macrophages to Salmonella Typhimurium infection. Cell Microbiol, 2016. 18(10): p. 1374-89. DOI: 10.1111/cmi.12578.
  48. Saha, S., et al., Leishmania donovani Exploits Macrophage Heme Oxygenase-1 To Neutralize Oxidative Burst and TLR Signaling-Dependent Host Defense. J Immunol, 2019. 202(3): p. 827-840. DOI: 10.4049/jimmunol.1800958.
  49. Zhang, Q., et al., HO-1 regulates the function of Treg: Association with the immune intolerance in vitiligo. J Cell Mol Med, 2018. 22(9): p. 4335-4343. DOI: 10.1111/jcmm.13723.
  50. Soares, M.P. and F.H. Bach, Heme oxygenase-1: from biology to therapeutic potential. Trends Mol Med, 2009. 15(2): p. 50-8. DOI: 10.1016/j.molmed.2008.12.004.
  51. Weiss, G., et al., Iron modulates interferon-gamma effects in the human myelomonocytic cell line THP-1. Exp Hematol, 1992. 20(5): p. 605-10.
  52. Oexle, H., et al., Pathways for the regulation of interferon-gamma-inducible genes by iron in human monocytic cells. J Leukoc Biol, 2003. 74(2): p. 287-94. DOI: 10.1189/jlb.0802420.
  53. Bogdan, C., Nitric oxide synthase in innate and adaptive immunity: an update. Trends Immunol, 2015. 36(3): p. 161-78. DOI: 10.1016/j.it.2015.01.003.
  54. Nairz, M., et al., Interferon-gamma limits the availability of iron for intramacrophage Salmonella typhimurium. Eur J Immunol, 2008. 38(7): p. 1923-36. DOI: 10.1002/eji.200738056.
  55. Nairz, M., et al., The co-ordinated regulation of iron homeostasis in murine macrophages limits the availability of iron for intracellular Salmonella typhimurium. Cell Microbiol, 2007. 9(9): p. 2126-40. DOI: 10.1111/j.1462-5822.2007.00942.x.
  56. Byrd, T.F., Tumor necrosis factor alpha (TNFalpha) promotes growth of virulent Mycobacterium tuberculosis in human monocytes iron-mediated growth suppression is correlated with decreased release of TNFalpha from iron-treated infected monocytes. J Clin Invest, 1997. 99(10): p. 2518-29. DOI: 10.1172/jci119436.
  57. Melillo, G., et al., Functional requirement of the hypoxia-responsive element in the activation of the inducible nitric oxide synthase promoter by the iron chelator desferrioxamine. J Biol Chem, 1997. 272(18): p. 12236-43. DOI: 10.1074/jbc.272.18.12236.
  58. Mabey, D.C., A. Brown, and B.M. Greenwood, Plasmodium falciparum malaria and Salmonella infections in Gambian children. J Infect Dis, 1987. 155(6): p. 1319-21. DOI: 10.1093/infdis/155.6.1319.
  59. Valdez, J.M., et al., Infections in patients with aplastic anemia. Semin Hematol, 2009. 46(3): p. 269-76. DOI: 10.1053/j.seminhematol.2009.03.008.
  60. Stefanova, D., et al., Hepcidin Protects against Lethal Escherichia coli Sepsis in Mice Inoculated with Isolates from Septic Patients. Infect Immun, 2018. 86(7). DOI: 10.1128/iai.00253-18.
  61. Camaschella, C., Iron and hepcidin: a story of recycling and balance. Hematology Am Soc Hematol Educ Program, 2013. 2013: p. 1-8. DOI: 10.1182/asheducation-2013.1.1.
  62. Fillebeen, C., et al., Hepcidin-mediated hypoferremic response to acute inflammation requires a threshold of Bmp6/Hjv/Smad signaling. Blood, 2018. 132(17): p. 1829-1841. DOI: 10.1182/blood-2018-03-841197.
  63. Barry, D.M. and A.W. Reeve, Increased incidence of gram-negative neonatal sepsis with intramuscula iron administration. Pediatrics, 1977. 60(6): p. 908-12.
  64. Christopher, G.W., Escherichia coli bacteremia, meningitis, and hemochromatosis. Arch Intern Med, 1985. 145(10): p. 1908.
  65. Robinson, A.E., J.R. Heffernan, and J.P. Henderson, The iron hand of uropathogenic Escherichia coli: the role of transition metal control in virulence. Future Microbiol, 2018. 13(7): p. 745-756. DOI: 10.2217/fmb-2017-0295.
  66. Shand, G.H., et al., In vivo evidence that bacteria in urinary tract infection grow under iron-restricted conditions. Infect Immun, 1985. 48(1): p. 35-9. DOI: 10.1128/iai.48.1.35-39.1985.
  67. Reddy, E.A., A.V. Shaw, and J.A. Crump, Community-acquired bloodstream infections in Africa: a systematic review and meta-analysis. Lancet Infect Dis, 2010. 10(6): p. 417-32. DOI: 10.1016/s1473-3099(10)70072-4.
  68. Berkley, J.A., et al., Bacteremia among children admitted to a rural hospital in Kenya. N Engl J Med, 2005. 352(1): p. 39-47. DOI: 10.1056/NEJMoa040275.
  69. Brent, A.J., et al., Incidence of clinically significant bacteraemia in children who present to hospital in Kenya: community-based observational study. Lancet, 2006. 367(9509): p. 482-8. DOI: 10.1016/s0140-6736(06)68180-4.
  70. Torres, V.J., et al., Staphylococcus aureus IsdB is a hemoglobin receptor required for heme iron utilization. J Bacteriol, 2006. 188(24): p. 8421-9. DOI: 10.1128/jb.01335-06.
  71. Haley, K.P. and E.P. Skaar, A battle for iron: host sequestration and Staphylococcus aureus acquisition. Microbes Infect, 2012. 14(3): p. 217-27. DOI: 10.1016/j.micinf.2011.11.001.

Reviewer 3 Report

The paper is interesting and well-organized.

However, it needs few improvements:

1) It would be pivotal to better characterize severe anaemia with respect to other types of anaemias, most of which presenting low serum iron levels. In this regard, the hematological parameters of African children affected by severe anaemia should be detailed and discussed.

2) The influence of inflammation on both the physiopathology of severe anaemia and the increased risk of bacterial infection should be included. In particular:

  • host immune response to pathogens’ infection is characterized by neutrophils recruitment and cytokines’ production. In the paper, it appears that immune response is always shut down in severe anaemia, even when in presence of pathogens’ infections. However, in Mandala et al. 2017, up-regulation of pro-inflammatory cytokines have been found in severe malarial anaemia. Please, re-organize the paragraph accordingly.
  • protracted inflammatory processes have been associated to intestinal barrier dysfunction and induction/maintenance of gut dysbiosis (Goethel et al., 2018). All these aspects should be discussed.
  • IL-6 has been widely reported as the main key cytokine in the regulation of systemic iron homeostasis, also in its combinatorial axis with hepcidin. Since hepcidin is low in severe anaemia, is there any evidence of the possible involvement of the sole IL-6?

Minor points:

1) The acronym for non-typhoid Salmonella, NTS, is first reported in line 37, however it is not applied in line 41 and again repeated in line 161. Please check throughout the manuscript.

2) Figure 2: commensals rather than commensals

Mandala et al. 2017 - Cytokine Profiles in Malawian Children Presenting with Uncomplicated Malaria, Severe Malarial Anemia, and Cerebral Malaria

Goethel et al. 2018 - The interplay between microbes and the immune response in inflammatory bowel disease

Author Response

The paper is interesting and well-organized.

However, it needs few improvements:

We thank the reviewer for their helpful comments, which we have addressed point-by-point below:

Point 1: It would be pivotal to better characterize severe anaemia with respect to other types of anaemias, most of which presenting low serum iron levels. In this regard, the hematological parameters of African children affected by severe anaemia should be detailed and discussed.

Response 1: Thank you for the suggestion. We have added the paragraph below in the introduction to highlight the different aetiologies of severe anaemia in African children and their haematological parameters.

Introduction, page 1, lines 40-63: “Severe anaemia aetiologies can be grouped into hyporegenerative (anaemias due to iron and other nutritional deficiencies, pure red cell aplasia, anaemia of inflammation, aplastic anaemia, erythropoietin underproduction and marrow infiltration) and regenerative (anaemias due to haemolysis, immune dysregulation, haemorrhage and non-immune factors [haemoglobinopathies, drugs, microangiopathy, and hypersplenism]) [1]. Absolute iron deficiency (defined as low serum iron, low ferritin, and elevated transferrin iron binding capacity) results from blood loss, increased physiological demands of iron, intake of staple foods with low iron bioavailability, and malabsorption. Diagnosis of iron deficiency anaemia is usually based on observation of microcytic red cell features, although haemoglobin E/C and alpha/beta thalassaemia need to be ruled out. Iron deficiency is a common cause of anaemia in sub-Saharan Africa [2], but is less frequently observed in children with severe anaemia [3-5]. Deficiencies of vitamin A and vitamin B12 are also common in African children [3], and are associated with severe anaemia [3, 5]. In sickle cell disease, anaemia is secondary to haemoglobin polymerisation leading to red blood cell deformation and lysis. Sickle cell disease may also induce iron deficiency anaemia through increased iron utilization to replace damaged red blood cells or urinary iron loss [6]. Anaemia of inflammation (low serum iron and normal/high ferritin levels) is found in patients with infections (parasitic, bacterial, viral, and fungal), cancer, or autoimmune disorders [1, 7], and is thought to be induced by the hepatic iron regulatory hormone hepcidin [8]. While persistently raised hepcidin levels may protect against invading pathogens [9, 10], iron sequestration may increase susceptibility of intracellular infections [11]. Regenerative anaemias are characterised by high reticulocyte counts due to increased haemolysis or haemorrhage [1]. In sub-Saharan Africa, little is known about severe anaemia aetiologies from a public health perspective. Nonetheless, most of these aetiologies are important, often coexist in a single patient [3, 5], and may contribute to risk of infection either individually or synergistically.

Point 2: The influence of inflammation on both the physiopathology of severe anaemia and the increased risk of bacterial infection should be included. In particular:

  1. host immune response to pathogens’ infection is characterized by neutrophils recruitment and cytokines’ production. In the paper, it appears that immune response is always shut down in severe anaemia, even when in presence of pathogens’ infections. However, in Mandala et al. 2017, up-regulation of pro-inflammatory cytokines have been found in severe malarial anaemia. Please, re-organize the paragraph accordingly..

Response 2a: Thank you for the suggestion. We have added more information on the influence of inflammation on the physiopathology of severe anaemia and increased risk of bacterial infection with particular focus on neutrophil and cytokine responses during severe anaemia in the paragraph below:

The severe anaemia – bacteraemia hypothesis, page 4, lines 187-205: “Protection against invasive bacterial infection relies on a coordinated and regulated innate and adaptive immune response. In the initial stages of infection, local macrophages engulf and destroy the invading pathogens, and produce cytokines and chemokines to induce an inflammatory response. Monocytes and neutrophils are rapidly recruited to the site of infection. These cells destroy phagocytosed organisms effectively using the NADPH oxidase-dependent ROS production [12, 13]. Neutrophils can also destroy extracellular organisms through degranulation, releasing the neutrophil-extracellular traps (NETs), or production of ROS [12, 14]. Severe anaemia due to iron deficiency may promote impaired development and apoptosis of immune cells [15-17], including neutrophil hypersegmentation [18] and impaired neutrophil/monocyte oxidative burst [19, 20]. There is limited clinical research regarding the effect of severe anaemia on white blood cell differential count. In some case reports, iron deficiency anaemia has been associated with neutropenia [21, 22]. The effects on neutrophils are supported by murine models of severe malaria anaemia, which reported reduced neutrophil influx and lower proinflammatory cytokine (IL-17 and IFN-?) levels [23, 24], and higher levels of the anti-inflammatory cytokine IL-10 (IL-10) [25, 26]. Elevated IL-6 levels have been observed in field studies of children with severe malaria anaemia [27, 28] and IL-6 may be upregulated when hepcidin levels are low [29] further inhibiting neutrophil influx [30]. IL-6 is an important checkpoint regulator of neutrophil trafficking and promotes clearance of neutrophils and recruitment of monocytes [30, 31]. Downregulation of neutrophil responses may contribute to poor clearance of invasive bacterial pathogens in children with severe anaemia.”

Point 2b: protracted inflammatory processes have been associated to intestinal barrier dysfunction and induction/maintenance of gut dysbiosis (Goethel et al., 2018). All these aspects should be discussed.

Response 2b: Thank you for the comments. We have included a discussion on how sustained inflammation may increase intestinal barrier dysfunction and gut dysbiosis as below:

The severe anaemia – bacteraemia hypothesis, page 4, lines 166-176: “In model studies, neonatal mice with severe anaemia had a persistent increase in intestinal permeability and electron micrographs showed abnormalities of epithelial adherens junctions probably due to destabilization of the E-cadherin mRNA [32] or decreased expression of the tight junction protein zonula occludens-1 (ZO-1) [33]. Phlebotomy-induced severe anaemia was associated with increased intestinal mucosal hypoxia and production of IFN-γ by intestinal macrophages, which may contribute to increased gut permeability and development of necrotizing enterocolitis [33]. Depletion of intestinal macrophages ablated the effects of severe anaemia on intestinal barrier activity. Whether other immune cells, including neutrophils [34], also contribute to such necrotizing enterocolitis remains unknown. Gut inflammation may also directly induce dysbiosis, through the actions of ROS, calprotectin, and other inflammatory mediators [35-37].”

Point 2c. IL-6 has been widely reported as the main key cytokine in the regulation of systemic iron homeostasis, also in its combinatorial axis with hepcidin. Since hepcidin is low in severe anaemia, is there any evidence of the possible involvement of the sole IL-6?

Response 2c: Thank you for the suggestion. We have now added discussions on how proinflammatory cytokines, including IL-6 may influence iron homeostasis and risk of invasive bacterial infections.

The severe anaemia – bacteraemia hypothesis, page 4, lines 144-155: “On the other hand, chronic infection may induce functional iron deficiency. Such “hypoferraemia of infection” is a host defence strategy to limit iron availability from invading pathogens [38], and involves sustained production of proinflammatory cytokines such as interferon-gamma (IFN-γ), tumour necrosis factor-alpha (TNF-α), interleukin (IL)-1 and IL-6 [7]. These proinflammatory cytokines can exacerbate existing severe anaemia by promoting dyserythropoiesis, iron sequestration and erythrophagocytosis [7, 39]. IL-6 promotes the production of hepcidin [8], and has been reported to be upregulated in children with severe malarial anaemia [27, 28]. However, ERFE may have a stronger negative effect on hepcidin production during severe anaemia [40], and low hepcidin levels have been observed in anaemic children and young women with concomitant inflammation [41, 42]. Whether and how IL-6 or other proinflammatory cytokines may increase iron sequestration independently of hepcidin in severely anaemic children remain unknown."

The severe anaemia – bacteraemia hypothesis, page 5, lines 200-205: “Elevated IL-6 levels have been observed in field studies of children with severe malarial anaemia [27, 28] and IL-6 may be upregulated when hepcidin levels are low [29] further inhibiting neutrophil influx [30]. IL-6 is an important checkpoint regulator of neutrophil trafficking and promotes clearance of neutrophils and recruitment of monocytes [30, 31]. Downregulation of neutrophil responses may contribute to poor clearance of invasive bacterial pathogens in children with severe anaemia.”

Minor points:

Point 1: The acronym for non-typhoid Salmonella, NTS, is first reported in line 37, however it is not applied in line 41 and again repeated in line 161. Please check throughout the manuscript.

Response 1: We have revised lines 66 and 277 from non-typhoidal Salmonella to NTS.

Point 2: Figure 2: commensals rather than commensals

 Response 2: We have now edited the figure to state commensals rather than comensals.

Mandala et al. 2017 - Cytokine Profiles in Malawian Children Presenting with Uncomplicated Malaria, Severe Malarial Anemia, and Cerebral Malaria

Goethel et al. 2018 - The interplay between microbes and the immune response in inflammatory bowel disease

References

  1. Lambert, J. and P. Beris, Pathophysiology and differential diagnosis of anaemia. Beaumont C, Béris P, Beuzard Y, Brugnara C. The Handbook on disorders of erythropoiesis, erythrocytes and iron metabolism. Paris: European school of haematology, 2009: p. 108-41.
  2. Kassebaum, N.J., The Global Burden of Anemia. Hematol Oncol Clin North Am, 2016. 30(2): p. 247-308. DOI: 10.1016/j.hoc.2015.11.002.
  3. Calis, J.C., et al., Severe anemia in Malawian children. N Engl J Med, 2008. 358(9): p. 888-99. DOI: 10.1056/NEJMoa072727.
  4. Foote, E.M., et al., Determinants of anemia among preschool children in rural, western Kenya. Am J Trop Med Hyg, 2013. 88(4): p. 757-64. DOI: 10.4269/ajtmh.12-0560.
  5. van Hensbroek, M.B., F. Jonker, and I. Bates, Severe acquired anaemia in Africa: new concepts. Br J Haematol, 2011. 154(6): p. 690-5. DOI: 10.1111/j.1365-2141.2011.08761.x.
  6. Sundd, P., M.T. Gladwin, and E.M. Novelli, Pathophysiology of Sickle Cell Disease. Annu Rev Pathol, 2019. 14: p. 263-292. DOI: 10.1146/annurev-pathmechdis-012418-012838.
  7. Nemeth, E. and T. Ganz, Anemia of inflammation. Hematol Oncol Clin North Am, 2014. 28(4): p. 671-81, vi. DOI: 10.1016/j.hoc.2014.04.005.
  8. Nemeth, E., et al., IL-6 mediates hypoferremia of inflammation by inducing the synthesis of the iron regulatory hormone hepcidin. Journal of Clinical Investigation, 2004. 113(9): p. 1271-1276. DOI: 10.1172/jci200420945.
  9. Arezes, J., et al., Hepcidin-induced hypoferremia is a critical host defense mechanism against the siderophilic bacterium Vibrio vulnificus. Cell Host Microbe, 2015. 17(1): p. 47-57. DOI: 10.1016/j.chom.2014.12.001.
  10. Wang, H.Z., et al., Hepcidin is regulated during blood-stage malaria and plays a protective role in malaria infection. The Journal of immunology, 2011. 187(12): p. 6410-6. DOI: 10.4049/jimmunol.1101436.
  11. Lokken, K.L., et al., Malaria Parasite-Mediated Alteration of Macrophage Function and Increased Iron Availability Predispose to Disseminated Nontyphoidal Salmonella Infection. Infect Immun, 2018. 86(9): p. e00301-18. DOI: 10.1128/iai.00301-18.
  12. Kruger, P., et al., Neutrophils: Between host defence, immune modulation, and tissue injury. PLoS Pathog, 2015. 11(3): p. e1004651. DOI: 10.1371/journal.ppat.1004651.
  13. Gogoi, M., M.M. Shreenivas, and D. Chakravortty, Hoodwinking the Big-Eater to Prosper: The Salmonella -Macrophage Paradigm. Journal of Innate Immunity, 2019. 11(3): p. 289-299. DOI: http://dx.doi.org/10.1159/000490953.
  14. Brinkmann, V., et al., Neutrophil extracellular traps kill bacteria. Science, 2004. 303(5663): p. 1532-5. DOI: 10.1126/science.1092385.
  15. Jiang, Y., et al., Iron-dependent histone 3 lysine 9 demethylation controls B cell proliferation and humoral immune responses. Nat Commun, 2019. 10(1): p. 2935. DOI: 10.1038/s41467-019-11002-5.
  16. Hassan, T.H., et al., Impact of iron deficiency anemia on the function of the immune system in children. Medicine (Baltimore), 2016. 95(47): p. e5395. DOI: 10.1097/md.0000000000005395.
  17. Das, I., et al., Impact of iron deficiency anemia on cell-mediated and humoral immunity in children: A case control study. J Nat Sci Biol Med, 2014. 5(1): p. 158-63. DOI: 10.4103/0976-9668.127317.
  18. Westerman, D.A., D. Evans, and J. Metz, Neutrophil hypersegmentation in iron deficiency anaemia: a case-control study. Br J Haematol, 1999. 107(3): p. 512-5. DOI: 10.1046/j.1365-2141.1999.01756.x.
  19. Berrak, S.G., et al., The effects of iron deficiency on neutrophil/monocyte apoptosis in children. Cell Prolif, 2007. 40(5): p. 741-54. DOI: 10.1111/j.1365-2184.2007.00460.x.
  20. Murakawa, H., et al., Iron deficiency and neutrophil function: different rates of correction of the depressions in oxidative burst and myeloperoxidase activity after iron treatment. Blood, 1987. 69(5): p. 1464-8.
  21. Abdelmahmuod, E., M.A. Yassin, and M. Ahmed, Iron Deficiency Anemia-Induced Neutropenia in Adult Female. Cureus, 2020. 12(6): p. e8899. DOI: 10.7759/cureus.8899.
  22. Abuirmeileh, A., A. Bahnassi, and A. Abuirmeileh, Unexplained chronic leukopenia treated with oral iron supplements. Int J Clin Pharm, 2014. 36(2): p. 264-7. DOI: 10.1007/s11096-013-9897-2.
  23. Mooney, J.P., et al., The mucosal inflammatory response to non-typhoidal Salmonella in the intestine is blunted by IL-10 during concurrent malaria parasite infection. Mucosal Immunol, 2014. 7(6): p. 1302-11. DOI: 10.1038/mi.2014.18.
  24. Mooney, J.P., L.J. Galloway, and E.M. Riley, Malaria, anemia, and invasive bacterial disease: A neutrophil problem? J Leukoc Biol, 2019. 105(4): p. 645-655. DOI: 10.1002/jlb.3ri1018-400r.
  25. Lokken, K.L., et al., Malaria parasite infection compromises control of concurrent systemic non-typhoidal Salmonella infection via IL-10-mediated alteration of myeloid cell function. PLoS Pathog, 2014. 10(5): p. e1004049. DOI: 10.1371/journal.ppat.1004049.
  26. Roux, C.M., et al., Both hemolytic anemia and malaria parasite-specific factors increase susceptibility to Nontyphoidal Salmonella enterica serovar typhimurium infection in mice. Infect Immun, 2010. 78(4): p. 1520-7. DOI: 10.1128/iai.00887-09.
  27. Mandala, W.L., et al., Cytokine Profiles in Malawian Children Presenting with Uncomplicated Malaria, Severe Malarial Anemia, and Cerebral Malaria. Clin Vaccine Immunol, 2017. 24(4). DOI: 10.1128/cvi.00533-16.
  28. Burte, F., et al., Circulatory hepcidin is associated with the anti-inflammatory response but not with iron or anemic status in childhood malaria. Blood, 2013. 121(15): p. 3016-22. DOI: 10.1182/blood-2012-10-461418.
  29. Pagani, A., et al., Low hepcidin accounts for the proinflammatory status associated with iron deficiency. Blood, 2011. 118(3): p. 736-46. DOI: 10.1182/blood-2011-02-337212.
  30. Kaplanski, G., et al., IL-6: a regulator of the transition from neutrophil to monocyte recruitment during inflammation. Trends Immunol, 2003. 24(1): p. 25-9. DOI: 10.1016/s1471-4906(02)00013-3.
  31. Fielding, C.A., et al., IL-6 regulates neutrophil trafficking during acute inflammation via STAT3. J Immunol, 2008. 181(3): p. 2189-95. DOI: 10.4049/jimmunol.181.3.2189.
  32. MohanKumar, K., et al., Severe neonatal anemia increases intestinal permeability by disrupting epithelial adherens junctions. Am J Physiol Gastrointest Liver Physiol, 2020. 318(4): p. G705-g716. DOI: 10.1152/ajpgi.00324.2019.
  33. Arthur, C.M., et al., Anemia induces gut inflammation and injury in an animal model of preterm infants. Transfusion, 2019. 59(4): p. 1233-1245. DOI: 10.1111/trf.15254.
  34. Spees, A.M., et al., Neutrophils are a source of gamma interferon during acute Salmonella enterica serovar Typhimurium colitis. Infect Immun, 2014. 82(4): p. 1692-7. DOI: 10.1128/iai.01508-13.
  35. Winter, S.E., et al., Gut inflammation provides a respiratory electron acceptor for Salmonella. Nature, 2010. 467(7314): p. 426-9. DOI: 10.1038/nature09415.
  36. Liu, J.Z., et al., Zinc sequestration by the neutrophil protein calprotectin enhances Salmonella growth in the inflamed gut. Cell Host Microbe, 2012. 11(3): p. 227-39. DOI: 10.1016/j.chom.2012.01.017.
  37. Goethel, A., K. Croitoru, and D.J. Philpott, The interplay between microbes and the immune response in inflammatory bowel disease. J Physiol, 2018. 596(17): p. 3869-3882. DOI: 10.1113/jp275396.
  38. Nairz, M., et al., Iron and innate antimicrobial immunity-Depriving the pathogen, defending the host. J Trace Elem Med Biol, 2018. 48: p. 118-133. DOI: 10.1016/j.jtemb.2018.03.007.
  39. de Bruin, A.M., C. Voermans, and M.A. Nolte, Impact of interferon-gamma on hematopoiesis. Blood, 2014. 124(16): p. 2479-86. DOI: 10.1182/blood-2014-04-568451.
  40. Latour, C., et al., Erythroferrone contributes to hepcidin repression in a mouse model of malarial anemia. Haematologica, 2017. 102(1): p. 60-68. DOI: 10.3324/haematol.2016.150227.
  41. Jonker, F.A., et al., Low hepcidin levels in severely anemic malawian children with high incidence of infectious diseases and bone marrow iron deficiency. PLoS One, 2013. 8(12): p. e78964. DOI: 10.1371/journal.pone.0078964.
  42. Stoffel, N.U., et al., The opposing effects of acute inflammation and iron deficiency anemia on serum hepcidin and iron absorption in young women. Haematologica, 2019. 104(6): p. 1143-1149. DOI: 10.3324/haematol.2018.208645.

Round 2

Reviewer 3 Report

The manuscript is greatly improved following reviewers' suggestions.